# Score-Based Diffusion meets Annealed Importance Sampling

**Arnaud Doucet, Will Grathwohl, Alexander G. D. G. Matthews & Heiko Strathmann** *
DeepMind
{arnauddoucet,wgrathwohl,alexmatthews,strathmann}@google.com

## Abstract

More than twenty years after its introduction, Annealed Importance Sampling (AIS) remains one of the most effective methods for marginal likelihood estimation. It relies on a sequence of distributions interpolating between a tractable initial distribution and the target distribution of interest which we simulate from approximately using a non-homogeneous Markov chain. To obtain an importance sampling estimate of the marginal likelihood, AIS introduces an extended target distribution to reweight the Markov chain proposal. While much effort has been devoted to improving the proposal distribution used by AIS, an underappreciated issue is that AIS uses a convenient but suboptimal extended target distribution. We here leverage recent progress in score-based generative modeling (SGM) to approximate the optimal extended target distribution minimizing the variance of the marginal likelihood estimate for AIS proposals corresponding to the discretization of Langevin and Hamiltonian dynamics. We demonstrate these novel, differentiable, AIS procedures on a number of synthetic benchmark distributions and variational auto-encoders.

## 1 Introduction

Evaluating the marginal likelihood, also known as evidence, is of key interest in Bayesian statistics as it allows not only model comparison but is also often used to select hyperparameters. A large variety of Monte Carlo methods have been proposed to address this problem, including path sampling [19], AIS [37] and related Sequential Monte Carlo methods [13]. An appealing feature of AIS is that it provides an unbiased estimate of the marginal likelihood and can thus be used to define an evidence lower bound (ELBO) or mutual information bounds; see e.g. [53, 51, 7].

AIS builds a proposal distribution using a Markov chain $(x_k)_{k=0}^K$ initialized at an easy-to-sample distribution followed by a sequence of Markov chain Monte Carlo (MCMC) transitions targeting typically annealed versions of the posterior. By proceeding this way, we obtain a proposal $x_K$ whose distribution is expected to be a reasonable approximation to the target posterior. However, this distribution is intractable as it requires integrating the joint proposal distribution over previous states $(x_k)_{k=0}^{K-1}$. AIS bypasses this issue by instead using Importance Sampling (IS) on the whole path $(x_k)_{k=0}^K$ through the introduction of an artificial extended target distribution whose marginal at time $K$ coincides with the posterior.

There has been much work devoted to improving AIS in machine learning and statistics but also in physics where it was introduced independently in [29, 9]. A standard approach to improve AIS is to modify the intermediate distributions [45, 21, 34] and corresponding transition kernels of the proposal [10, 53, 18, 51, 54]. We here address a distinct problem. For a given proposal, it was shown in [13] that the extended target distribution minimizing the variance of the evidence estimate is not

---

*alphabetical order, equal contribution.

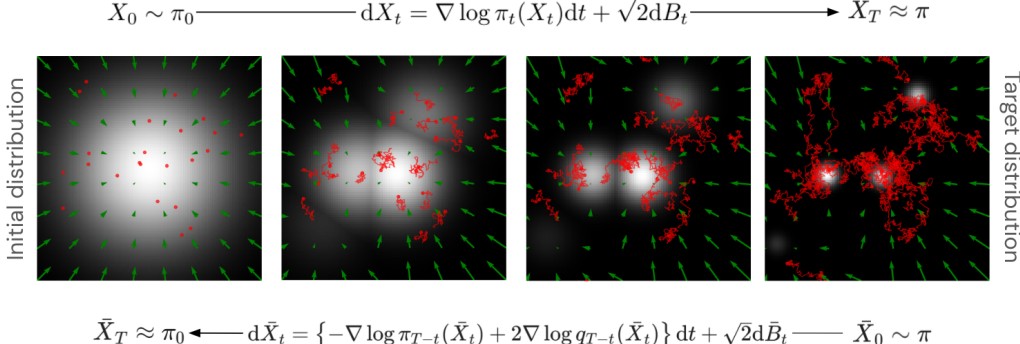

$$X_0 \sim \pi_0 \text{\textemdash} \mathrm{d}X_t = \nabla \log \pi_t(X_t)\mathrm{d}t + \sqrt{2}\mathrm{d}B_t \longrightarrow X_T \approx \pi$$

Initial distribution

Target distribution

$$\bar{X}_T \approx \pi_0 \longleftarrow \mathrm{d}\bar{X}_t = \left\{ -\nabla \log \pi_{T-t}(\bar{X}_t) + 2\nabla \log q_{T-t}(\bar{X}_t) \right\} \mathrm{d}t + \sqrt{2}\mathrm{d}\bar{B}_t \text{\textemdash} \bar{X}_0 \sim \pi$$

Figure 1: **Top**: Samples $X_t$ from an AIS proposal (red) obtained by sampling initially from a Gaussian at $t = 0$ and diffusing through Langevin dynamics on intermediate targets $\pi_t$ (white). The intermediate marginals of the proposal, $q_t$, approximated by the samples are such that $q_T \approx \pi$ for a reasonably fast mixing diffusion. **Bottom**: Computing importance weights. The optimal extended target used to compute the weights is the distribution obtained by initializing $\bar{X}_0$ exactly from $\pi$ and then following the reverse-time dynamics of the forward AIS proposal. This requires access to score vectors of the marginals $q_t$.

the one used by AIS but is instead defined through the time-reversal of the proposal. However, this result is difficult to exploit algorithmically as the time-reversal is intractable for useful proposals.

In this paper, we show how one can combine this result with recent advances in SGM to obtain improved, lower variance, AIS estimates. We concentrate on scenarios where we use unadjusted overdamped Langevin [23, 53, 51] and unadjusted Hamiltonian proposals with partial momentum refreshment (i.e. underdamped Langevin) [10, 53, 18, 54, 28] which correspond to time-discretized diffusion processes. The first benefit of using such proposals is that, by omitting Metropolis–Hastings steps, one obtains differentiable versions of the Evidence Lower Bound (ELBO) amenable to the reparameterization trick. The second benefit of these proposals is that their time-reversal can be approximated by adapting techniques developed for SGM [24, 48, 15] to our setup. We derive a principled parameterization for an approximation of their time-reversal which we learn by maximizing the ELBO. As for SGM, this ELBO coincides with a denoising score matching loss [27, 52, 24, 48]. This provides novel, optimized and differentiable, AIS estimators which we refer to as Monte Carlo Diffusion (MCD). We demonstrate the benefits of this approach on synthetic benchmark distributions and variational auto-encoders (VAEs) [31]. All proofs can be found in the Appendix. A preliminary version of this work appeared in [16].

## 2 Annealed Importance Sampling

### 2.1 Setup and algorithm

Consider a probability density $\pi$ on $\mathbb{R}^d$ of the form

$$\pi(x) = \frac{\gamma(x)}{Z}, \qquad Z = \int_{\mathbb{R}^d} \gamma(x)\mathrm{d}x, \tag{1}$$

where $\gamma(x)$ can be evaluated pointwise. We want to approximate the intractable normalizing constant $Z$. In a Bayesian framework, $\gamma(x) = p(x)p(\mathcal{D}|x)$ is the joint density of parameter $x$ and data $\mathcal{D}$, $\pi(x) = p(x|\mathcal{D})$ the corresponding posterior and $Z = p(\mathcal{D})$ the evidence.

To estimate $Z$, AIS introduces the intermediate distributions $(\pi_k)_{k=1}^K$ bridging smoothly from a tractable distribution $\pi_0$ to the target distribution $\pi_K = \pi$ of interest. One typically uses $\pi_k(x) \propto \gamma_k(x)$ with $\gamma_k(x) = \pi_0(x)^{1-\beta_k}\gamma(x)^{\beta_k}$ for $0 = \beta_0 < \beta_1 < \cdots < \beta_K = 1$ but other choices are possible [21]. The IS proposal used by AIS is then obtained by running a Markov chain $(x_k)_{k=0}^K$ such that $x_0 \sim \pi_0(\cdot)$, and then $x_k \sim F_k(\cdot|x_{k-1})$ for $k \geq 1$ where $F_k$ is a MCMC kernel invariant w.r.t. $\pi_k$. The proposal is thus given by

$$Q(x_{0:K}) = \pi_0(x_0) \prod_{k=1}^K F_k(x_k|x_{k-1}). \tag{2}$$

Denote by $q_k$ the marginal distribution of $x_k$ under $Q$ satisfying $q_k(x_k) = \int q_{k-1}(x_{k-1})F_k(x_k|x_{k-1})\mathrm{d}x_{k-1}$ for $k \geq 1$ and $q_0 = \pi_0$, it is typically intractable for $k \geq 1$. As $q_K$ cannot be evaluated in complex scenarios, the marginal IS estimate $w_{\mathrm{mar}}(x_K) = \gamma(x_K)/q_K(x_K)$ of $Z$ is intractable.

One can bypass this issue by introducing an extended target distribution

$$P(x_{0:K}) = \frac{\Gamma(x_{0:K})}{Z}, \qquad \Gamma(x_{0:K}) = \gamma(x_K)\prod_{k=0}^{K-1} B_k(x_k|x_{k+1}), \tag{3}$$

where $(B_k)_{k=0}^{K-1}$ are backward Markov transition kernels, i.e. $\int B_k(x_k|x_{k+1})\mathrm{d}x_k = 1$ for any $x_{k+1}$, so that by construction $x_K \sim \pi$ under $P$. For any selection of backward kernels such that the ratio $\Gamma/Q$ is well-defined, we then have

$$\mathbb{E}_Q[w(x_{0:K})] = Z, \quad \text{for } w(x_{0:K}) = \frac{\Gamma(x_{0:K})}{Q(x_{0:K})}, \tag{4}$$

i.e. $w(x_{0:K})$ is an unbiased estimate of $Z$ for $x_{0:K} \sim Q$.

The AIS estimate of the evidence is a specific instance of the estimator (4) relying on the backward kernels $B_k^{\mathrm{ais}}(x_k|x_{k+1}) = \pi_{k+1}(x_k)F_{k+1}(x_{k+1}|x_k)/\pi_{k+1}(x_{k+1})$. This yields the following expression for $\log w(x_{0:K})$:

$$\log w_{\mathrm{ais}}(x_{0:K}) = \textstyle\sum_{k=1}^{K} \log\big(\gamma_k(x_{k-1})/\gamma_{k-1}(x_{k-1})\big). \tag{5}$$

## 2.2 Limitations of AIS

While designing $P$ in (3) by using the backward Markov kernels $(B_k^{\mathrm{ais}})_{k=0}^{K-1}$ is convenient, it is also suboptimal in terms of variance. For example, consider the ideal scenario where $F_k(x_k|x_{k-1}) = \pi_k(x_k)$. This scenario has been used many times in the literature to provide some guidelines on AIS, see e.g. [37, 21]. In this case, $\mathrm{var}_Q[\log w_{\mathrm{ais}}(x_{0:K})] = \sum_{k=1}^{K} \mathrm{var}_{\pi_{k-1}}[\log(\gamma_k(x_{k-1})/\gamma_{k-1}(x_{k-1}))] > 0$ while $\mathrm{var}_{q_K}[w_{\mathrm{mar}}(x_K)] = \mathrm{var}_\pi[w_{\mathrm{mar}}(x_K)] = 0$.

Another illustration of the suboptimality of AIS is to consider a scenario where the proposal is a homogeneous MCMC chain, i.e. $x_0 \sim \pi_0$ and $x_k \sim F(\cdot|x_{k-1})$ for $F$ a $\pi$-invariant MCMC kernel; i.e. use $F_k = F$ and $\pi_k = \pi$ for $k = 1, ..., K$. If $F$ is reasonably well-mixing, then $q_K \approx \pi$ for $K$ large enough and the evidence estimate $w_{\mathrm{mar}}(x_K) = \gamma(x_K)/q_K(x_K)$ should have small variance. However, it is easy to check that we have $w_{\mathrm{ais}}(x_{0:K}) = \gamma(x_0)/\pi_0(x_0)$ for the exact same proposal; i.e. the AIS estimate does not depend on the MCMC samples $x_{1:K}$ and boils down to the IS estimate of $Z$ using the proposal $\pi_0$.

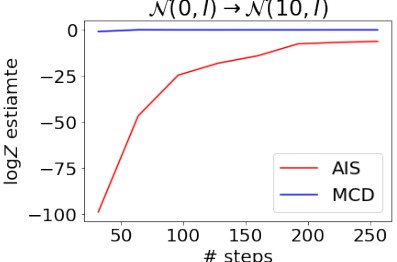

Figure 2: Comparing $\log Z$ estimates as a function of $K$ using AIS and MCD. Both estimates use the same forward kernels but reweight samples in a different way using distinct backward kernels. Initial distribution $\pi_0$ is 20-dimensional $\mathcal{N}(0, I)$ and progressively shifts to the target $\pi = \pi_K = \mathcal{N}(10, I)$. The MCD estimate is much closer to the ground truth ($\log Z = 0$) than AIS.

These two examples illustrate that it would be preferable to use $w_{\mathrm{mar}}(x_K)$ rather than $w_{\mathrm{ais}}(x_{0:K})$. In Appendix A, we provide a detailed comparison of both estimates in a scenario where their variance can be computed analytically. We propose in the next section an unbiased estimate of the evidence (MCD) approximating $w_{\mathrm{mar}}(x_K)$ based on a different choice of backward kernels. As illustrated in Figure 2, significant gains can be achieved.

## 3 Optimized Annealed Importance Sampling

We show here that the optimal extended target distribution $P$ minimizing the variance of the evidence estimate (4) is defined through the time-reversal of the proposal $Q$. By exploiting a connection to SGM, we can approximate this reversal using score matching when the proposal is obtained through an unadjusted overdamped or underdamped Langevin algorithm.

## 3.1 Optimal Extended Target Distribution via Time Reversal

We summarize here Proposition 1 of [13]; see also [46, 5].

**Proposition 1.** *For a proposal $Q$ of the form (2), the extended target $P$ of the form (3) minimizing both the Kullback–Leibler divergence $D_{\mathrm{KL}}(Q||P)$ and the variance of the evidence estimate $w(x_{0:K}) = \Gamma(x_{0:K})/Q(x_{0:K})$ for $x_{0:K} \sim Q$ is given by $P^{\mathrm{opt}}(x_{0:K}) = \Gamma^{\mathrm{opt}}(x_{0:K})/Z$ where*

$$\Gamma^{\mathrm{opt}}(x_{0:K}) = \gamma(x_K) \prod_{k=0}^{K-1} B_k^{\mathrm{opt}}(x_k|x_{k+1}), \qquad B_k^{\mathrm{opt}}(x_k|x_{k+1}) = \frac{q_k(x_k)F_{k+1}(x_{k+1}|x_k)}{q_{k+1}(x_{k+1})}. \quad (6)$$

*In particular, one has*

$$w_{\mathrm{mar}}(x_K) = \frac{\gamma(x_K)}{q_K(x_K)} = \frac{\Gamma^{\mathrm{opt}}(x_{0:K})}{Q(x_{0:K})}, \quad and \quad D_{\mathrm{KL}}(Q||P^{\mathrm{opt}}) = D_{\mathrm{KL}}(q_K||\pi). \quad (7)$$

This result follows simply from the chain rule and the law of total variance which yield

$$D_{\mathrm{KL}}(Q||P) = D_{\mathrm{KL}}(q_K||\pi) + \mathbb{E}_{q_K}\Big[D_{\mathrm{KL}}(Q(\cdot|x_K)||P(\cdot|x_K))\Big], \quad (8)$$

$$\mathrm{var}_Q[w(x_{0:K})] = \mathrm{var}_{q_K}[w_{\mathrm{mar}}(x_K)] + \mathbb{E}_{q_K}[\mathrm{var}_{Q(\cdot|x_K)}[w(x_{0:K})]]. \quad (9)$$

Both quantities are clearly minimized by selecting $P(x_{0:K-1}|x_K) = Q(x_{0:K-1}|x_K)$.

We emphasize that Proposition 1 applies to any forward kernels $(F_k)_{k=1}^K$ including MCMC kernels, unadjusted Langevin kernels or even deterministic maps[2]. It shows that $P^{\mathrm{opt}}$ is the distribution of a backward process initialized at $\pi$ which then follows the time-reversed dynamics of the forward process $Q$. If we had $q_K = \pi$, then we would have $P^{\mathrm{opt}} = Q$ as then $P^{\mathrm{opt}}$ would correspond to the backward decomposition of $Q$.

## 3.2 Time reversal, Score matching and ELBO for unadjusted overdamped Langevin

We concentrate here on the case where $(F_k)_{k=1}^K$ correspond to a time-inhomogeneous unadjusted (overdamped) Langevin algorithm (ULA) as used in [23, 53, 51]; that is we consider $F_k(x_k|x_{k-1}) = \mathcal{N}(x_k; x_{k-1} + \delta\nabla\log\pi_k(x_{k-1}), 2\delta I)$ where $\delta > 0$ is a stepsize. Let $\delta := T/K$ then, as $K \to \infty$, the proposal $Q$ converges to the path measure $\mathcal{Q}$ of the following inhomogeneous Langevin diffusion $(x_t)_{t\in[0,T]}$ defined by the stochastic differential equation (SDE)

$$\mathrm{d}x_t = \nabla\log\pi_t(x_t)\mathrm{d}t + \sqrt{2}\mathrm{d}B_t, \qquad x_0 \sim \pi_0, \quad (10)$$

where $(B_t)_{t\in[0,T]}$ is standard multivariate Brownian motion and we are slightly abusing notation from now on as $\pi_t$ for $t = t_k = k\delta$ corresponds to $\pi_k$ in discrete-time. Many quantitative results measuring the discrepancy between the law of $x_T$ and $\pi_T = \pi$ for such annealed diffusions have been obtained; see e.g. [17, 50]. From [22], it is known that the time-reversed process $(\bar{x}_t) = (x_{T-t})_{t\in[0,T]}$ is also a diffusion given by

$$\mathrm{d}\bar{x}_t = \big\{-\nabla\log\pi_{T-t}(\bar{x}_t) + 2\nabla\log q_{T-t}(\bar{x}_t)\big\}\mathrm{d}t + \sqrt{2}\mathrm{d}\bar{B}_t, \qquad \bar{x}_0 \sim q_T, \quad (11)$$

where $(\bar{B}_t)_{t\in[0,T]}$ is another multivariate Brownian motion. The continuous-time version of $P^{\mathrm{opt}}$ is the path measure $\mathcal{P}^{\mathrm{opt}}$ defined by the diffusion (11) but initialized at $\bar{x}_0 \sim \pi$ rather than $q_T$ as noted in [5]; see Figure 1 for an illustration. This shows that approximating $(B_k^{\mathrm{opt}})_{k=0}^{K-1}$ requires approximating the so-called scores $(\nabla\log q_t(x))_{t\in[0,T]}$. This can be derived heuristically through the fact that a Taylor expansion yields the following approximation of the optimal backward kernels, $B_k^{\mathrm{opt}}(x_k|x_{k+1}) \approx \mathcal{N}(x_k; x_{k+1} - \delta\nabla\log\pi_{k+1}(x_{k+1}) + 2\delta\nabla\log q_{k+1}(x_{k+1}), 2\delta I)$, which indeed corresponds to a Euler discretization of (11); see e.g. [12, Section 2.1].

In SGM [48], one gradually adds noise to data using an Ornstein–Uhlenbeck diffusion to transform the complex data distribution into a Gaussian distribution and the generative model is obtained by approximating the time-reversal of this diffusion initialized by Gaussian noise. Practically, the time-reversal approximation is obtained by estimating the scores of the noising diffusion using denoising

---

[2]The case of deterministic maps corresponds to normalizing flow components, where the inverse flow is the optimal and only valid reversal, see e.g. [2] which includes a detailed literature review.

score matching [52]. While in our setup, the diffusion (10) instead goes from a simple distribution to a complex one (see Appendix D for a discussion), we can still use score matching ideas. We define a path measure $\mathcal{P}_\theta$ approximating $\mathcal{P}^{\mathrm{opt}}$ using a neural network $s_\theta(T-t, \bar{x}_t)$ in place of $\nabla \log q_{T-t}(\bar{x}_t)$ in (11), i.e. we consider

$$d\bar{x}_t = \big\{ -\nabla \log \pi_{T-t}(\bar{x}_t) + 2s_\theta(T-t, \bar{x}_t) \big\} dt + \sqrt{2} d\bar{B}_t, \qquad \bar{x}_0 \sim \pi. \tag{12}$$

We would like to learn $\theta$ by minimizing $D_{\mathrm{KL}}(\mathcal{Q}||\mathcal{P}_\theta)$ over $\theta$, i.e. equivalently we maximize a continuous-time ELBO. Note that it is neither easily feasible to minimize $D_{\mathrm{KL}}(\mathcal{P}^{\mathrm{opt}}||\mathcal{P}_\theta)$ (as one cannot sample from $\pi$) nor it is desirable as the evidence estimate is computed using samples from $\mathcal{Q}$. Hence we want the scores to be well-approximated in regions of high-probability mass under $\mathcal{Q}$.

In practice, the diffusions corresponding to $\mathcal{Q}$ and $\mathcal{P}_\theta$ have to be discretized, so a more direct route adopted here is to simply take inspiration from (11) and to consider the parameterized backward kernels $B_k^\theta(x_k|x_{k+1}) = \mathcal{N}(x_k; x_{k+1} - \delta \nabla \log \pi_{k+1}(x_{k+1}) + 2\delta s_\theta(k+1, x_{k+1}), 2\delta I)$ to obtain a parameterized extended target $P_\theta$ and corresponding unnormalized target $\Gamma_\theta$. We then learn $\theta$ by minimizing $D_{\mathrm{KL}}(Q||P_\theta)$ where

$$Q(x_{0:K}) = \pi_0(x_0) \prod_{k=0}^{K-1} F_{k+1}(x_{k+1}|x_k), \quad P_\theta(x_{0:K}) = \pi(x_K) \prod_{k=0}^{K-1} B_k^\theta(x_k|x_{k+1}).$$

This is obviously equivalent to maximizing the ELBO $\mathbb{E}_Q[\log w_\theta(x_{0:K})]$ where $w_\theta(x_{0:K}) = \Gamma_\theta(x_{0:K})/Q(x_{0:K})$. We note that it has previously been proposed to learn parameterized backward kernels for general AIS proposals [44, 26]. However, the parameterization adopted therein, $B_k^\theta(x_k|x_{k+1}) = \mathcal{N}(x_k; \mu_\theta(x_{k+1}), \Sigma_\theta(x_{k+1}))$, does not leverage the structure of the true reversal and performs poorly experimentally [51, Section 4.2].

As established in the next proposition, the continuous and discrete time approaches coincide for $\delta \ll 1$. Once $\theta$ is learned, we then obtain an unbiased estimate of $Z$ through $w_\theta(x_{0:K})$ for $x_{0:K} \sim Q$.

**Proposition 2.** *Under regularity conditions, we have*

$$D_{\mathrm{KL}}(\mathcal{Q}||\mathcal{P}_\theta) = \mathbb{E}_\mathcal{Q} \Big[ \int_0^T ||s_\theta(t, x_t) - \nabla \log q_t(x_t)||^2 dt \Big] + C_1 \tag{13}$$

$$= \sum_{k=1}^K \int_{t_{k-1}}^{t_k} \mathbb{E}_\mathcal{Q} \big[ ||s_\theta(t, x_t) - \nabla \log q_{t|t_{k-1}}(x_t|x_{t_{k-1}})||^2 \big] dt + C_2, \tag{14}$$

*where $t_k = k\delta$, $K = T/\delta$, $q_{t|s}(x'|x)$ is the density of $x_t = x'$ given $x_s = x$ under $\mathcal{Q}$ and $C_1, C_2$ constants independent of $\theta$. Let $\mathcal{L}(\theta) = \delta \sum_{k=1}^K \mathbb{E}_Q \big[ ||s_\theta(k, x_k) - \nabla \log F_k(x_k|x_{k-1})||^2 \big]$ denote a discrete-time approximation of this loss. We have $\nabla D_{\mathrm{KL}}(Q||P_\theta) = \nabla \mathcal{L}(\theta) + \epsilon(\theta)$ for some function $\epsilon$ satisfying $\lim_{K\to\infty} \epsilon(\theta) = 0$.*

Equation (13) shows that $D_{\mathrm{KL}}(\mathcal{Q}||\mathcal{P}_\theta)$ corresponds to a score matching loss as for SGM [47]. It is possible to rewrite this loss as (14) so as to replace the intractable score term $\nabla \log q_t(x_t)$ by the easy to approximate gradients of the log-transitions $\nabla \log q_t(x_t|x_{t_{k-1}})$ [52]. In practice, as mentioned above, we simply learn $\theta$ by minimizing the discrete-time KL discrepancy $D_{\mathrm{KL}}(Q||P_\theta)$. This formulation is also very convenient as we can additionally learn potential parameters $\phi$ of a $Q_\phi$ using the same criterion. Note from equation (8) that the KL divergence decomposes as firstly a term penalizing the difference in the approximating measure and the fixed target at the final time and secondly another term which can be reduced by optimization of both $Q_\phi$ and $P_\theta$ conditioned on $x_T$; see e.g. [1].

Pseudo-code for our approach (with a comparison to the AIS algorithm proposed in [23, 53, 51]) can be found in Algorithm 1.

## 3.3 Incorporating Hamiltonian dynamics via the underdamped Langevin equation

We now consider a proposal defined on an extended space which arises from the time-discretization of a time-inhomogeneous underdamped Langevin dynamics; see e.g. [33, Chapter 6]. In this scenario, we first focus on continuous-time as the development of suitable numerical integrators is much more involved than for overdamped Langevin diffusions. We consider the diffusion $(x_t, p_t)_{t \in [0,T]}$ where $p_t \in \mathbb{R}^d$ is a momentum variable

$$dx_t = M^{-1} p_t dt, \qquad dp_t = \nabla \log \pi_t(x_t) dt - \zeta p_t dt + \sqrt{2\zeta} M^{1/2} dB_t \tag{15}$$

---

**Algorithm 1** Unadjusted Langevin AIS/MCD – red instructions for AIS and blue for MCD

---

**Require:** Unnormalized target $\gamma(x)$, initial state proposal $\pi_0(x)$, number steps $K$, stepsize $\delta$, annealing schedule $\{\beta_k\}_{k=0}^K$, score model $s_\theta(k, x)$

Sample $x_0 \sim \pi_0(x_0)$

Set $\log w = -\log \pi_0(x_0)$

**for** $k = 1$ to $K$ **do**

    Define $\log \gamma_k(\cdot) = \beta_k \log \gamma(\cdot) + (1 - \beta_k) \log \pi_0(\cdot)$

    Define $F_k(x_k|x_{k-1}) = \mathcal{N}(x_k; x_{k-1} + \delta \nabla \log \gamma_k(x_{k-1}), 2\delta I)$

    Sample $x_k \sim F_k(\cdot|x_{k-1})$

    Define $B_{k-1}(x_{k-1}|x_k) = F_k(x_{k-1}|x_k)$                           ▷ AIS

    Define $B_{k-1}(x_{k-1}|x_k) = \mathcal{N}(x_{k-1}; x_k - \delta \nabla \log \gamma_k(x_k) + 2\delta s_\theta(k, x_k), 2\delta I)$    ▷ MCD

    Set $\log w = \log w + \log B_{k-1}(x_{k-1}|x_k) - \log F_k(x_k|x_{k-1})$

**end for**

Set $\log w = \log w + \log \gamma(x_K)$

---

initialized at $x_0 \sim \pi_0, p_0 \sim \mathcal{N}(0, M)$ defining the path measure $\mathcal{Q}$. Here $M$ is a positive definite mass matrix, $\zeta > 0$ a friction coefficient and $(B_t)_{t \in [0,T]}$ a multivariate Brownian motion. If $\pi_t$ was not time-varying, e.g. $\pi_t = \pi$, the invariant distribution of this diffusion would be given by $\bar{\pi}(x, p) = \pi(x)\mathcal{N}(p; 0, M)$. Intuitively, in the time varying case the SDE will have enough time to approximate each intermediate $\bar{\pi}_t(x, p) = \pi_t(x)\mathcal{N}(p; 0, M)$ if we change the target sufficiently slowly. We can think of underdamped Langevin as a continuous-time version of Hamiltonian dynamics with continuous stochastic partial momentum refreshment [25].

From [22], the time-reversal of the diffusion (15) is also a diffusion process $(\bar{x}_t, \bar{p}_t)_{t \in [0,T]} = (x_{T-t}, p_{T-t})_{t \in [0,T]}$ given by $(\bar{x}_0, \bar{p}_0) \sim \eta_T$ and

$$d\bar{x}_t = -M^{-1}\bar{p}_t dt, \tag{16}$$

$$d\bar{p}_t = -\nabla \log \pi_{T-t}(\bar{x}_t)dt + \zeta\bar{p}_t dt + 2\zeta M \nabla_{\bar{p}_t} \log \eta_{T-t}(\bar{x}_t, \bar{p}_t)dt + \sqrt{2\zeta}M^{1/2}d\bar{B}_t,$$

where $\eta_t$ denotes the density $(x_t, p_t)$ under (15). In this case, the continuous-time version of $P_{\text{opt}}$ is the path measure $\mathcal{P}_{\text{opt}}$ defined by the diffusion (16) but initialized at $\bar{x}_0 \sim \pi, \bar{p}_0 \sim \mathcal{N}(0, M)$ rather than $\eta_T$. We will approximate it by the path measure $\mathcal{P}_\theta$ using a neural network $s_\theta(T - t, \bar{x}_t, \bar{p}_t)$ in place of $\nabla \log \eta_{T-t}(\bar{x}_t, \bar{p}_t)$ in (16), i.e. we consider

$$d\bar{x}_t = -M^{-1}\bar{p}_t dt, \tag{17}$$

$$d\bar{p}_t = -\nabla \log \pi_{T-t}(\bar{x}_t)dt + \zeta\bar{p}_t dt + 2\zeta M s_\theta(T - t, \bar{x}_t, \bar{p}_t)dt + \sqrt{2\zeta}M^{1/2}d\bar{B}_t.$$

As for overdamped Langevin, we could also learn $\theta$ by minimizing $D_{\text{KL}}(\mathcal{Q}||\mathcal{P}_\theta)$ over $\theta$. This again corresponds to minimizing a score matching loss albeit of a form slightly different from (13).

**Proposition 3.** *Under regularity conditions, we have*

$$D_{\text{KL}}(\mathcal{Q}||\mathcal{P}_\theta) = \zeta \mathbb{E}_\mathcal{Q}\Big[ \int_0^T ||s_\theta(t, x_t, p_t) - \nabla_{p_t} \log \eta_t(x_t, p_t)||^2 dt \Big] + C_1 \tag{18}$$

$$= \zeta \sum_{k=1}^K \int_{t_{k-1}}^{t_k} \mathbb{E}_\mathcal{Q}\big[ ||s_\theta(t, x_t, p_t) - \nabla_{p_t} \log \eta_{t|t_{k-1}}(x_t, p_t|x_{t_{k-1}}, p_{t_{k-1}})||_M^2 \big] dt + C_2,$$

*where* $||x||_M := u^{\mathrm{T}}Mu$, $t_k = k\delta$, $K = T/\delta$, $C_1, C_2$ *are constants independent of* $\theta$ *and* $\eta_{t|s}(x', p'|x, p)$ *is the density of* $(x_t, p_t) = (x', p')$ *given* $(x_s, p_s) = (x, p)$ *under* $\mathcal{Q}$.

While the continuous-time perspective shed light on how to parameterize an approximation to the time-reversal, this does not lead directly to an implementable discrete-time algorithm for underdamped Langevin. Contrary to overdamped Langevin, we cannot indeed simply use an Euler discretization of (15) defining $\mathcal{Q}$ and (17) defining $\mathcal{P}_\theta$ to obtain some discrete-time forward and backward kernels and then compute $w_\theta(x_{0:K}, p_{0:K}) = \Gamma_\theta(x_{0:K}, p_{0:K})/Q(x_{0:K}, p_{0:K})$. This is because this ratio is not well-defined due to the lack of noise on the position component in both (15) and (17).

The integrator we use for the forward equation (15), consists in alternating partial momentum refreshments and deterministic leapfrog steps (see e.g [33, 38]) giving

$$\tilde{p}_{t+\delta} \sim \mathcal{N}(hp_t, (1 - h^2)M), \qquad (x_{t+\delta}, p_{t+\delta}) = \Phi_t(x_t, \tilde{p}_{t+\delta}), \tag{19}$$

---

**Algorithm 2** Unadjusted Hamiltonian AIS/MCD – red instructions for AIS and blue for MCD

---

**Require:** Unnormalized target $\gamma(x)$, initial state proposal $\pi_0(x)$, number steps $K$, stepsize $\eta$, annealing schedule $\{\beta_k\}_{k=0}^K$, damping coefficient $h$, mass matrix $M$, score model $s_\theta(k, x, p)$

    Sample $x_0 \sim \pi_0(x_0)$ and $p_0 \sim \mathcal{N}(p_0; 0, M)$

    Set $\log w = -\log \pi_0(x_0) - \log \mathcal{N}(p_0; 0, M)$

    **for** $k = 1$ to $K$ **do**

        Define $\log \gamma_k(\cdot) = \beta_k \log \gamma(\cdot) + (1 - \beta_k) \log \pi_0(\cdot)$

        Sample $\tilde{p}_k \sim \mathcal{N}(hp_{k-1}, (1 - h^2)M)$

        Set $\mu_q = p_{k-1}$

        Set $\mu_p = \tilde{p}_k$                                              ▷ UHA reversal mean

        Set $\mu_p = \tilde{p}_k - 2\log(h)[Ms_\theta(k, x_{k-1}, \tilde{p}_k) + \tilde{p}_k]$       ▷ MCD reversal mean

        Set $\log w = \log w + \log \mathcal{N}(p_{k-1}; h\mu_p, (1 - h)^2 M) - \log \mathcal{N}(\tilde{p}_k; h\mu_q, (1 - h)^2 M)$

        Run leapfrog integrator on $\gamma_k$ and set $(x_k, p_k) = \Phi(x_{k-1}, \tilde{p}_k)$

    **end for**

    Set $\log w = \log w + \log \gamma(x_K) + \log \mathcal{N}(p_K; 0, M)$

---

with $h = \exp\{-\zeta\delta\}$ and $\Phi_t$ is the leapfrog integrator for $\pi_t$. The resulting forward sampler is similar to the one proposed by [18], except we do not flip the momentum after the leapfrog step[3]. This integrator may be interpreted as a splitting method for equation (15); see e.g [33, Chapter 7].

We need the integrator for the reversal to fulfill two criteria. First, by definition, as the time step $\delta \to 0$ it must recover the SDE (17). Second, the importance weight of the forward sampler to the reversal must be well defined. Since the leapfrog integrator is a diffeomorphism (or flow) the only possible way to get a well defined reversal for these steps is to take the inverse $\Phi_t^{-1}$. As the transformation is also volume preserving, the contribution from the deterministic forward and reverse terms will then exactly cancel in the importance weight. The required form of the reverse integrator is

$$(x_t, \tilde{p}_{t+\delta}) = \Phi_t^{-1}(x_{t+\delta}, p_{t+\delta}), \qquad p_t \sim \mathcal{N}(hf_\theta(t + \delta, x_t, \tilde{p}_{t+\delta}), (1 - h^2)M), \qquad (20)$$

where $f_\theta(t + \delta, x_t, \tilde{p}_{t+\delta}) := \tilde{p}_{t+\delta} + \delta 2\zeta[Ms_\theta(t, x_t, \tilde{p}_{t+\delta}) + \tilde{p}_{t+\delta}]$. In Appendix C we show that as $\delta \to 0$ this can indeed be interpreted as a valid split integrator for the reverse SDE (17). The crucial point of algorithmic difference from [10, 18, 54] arises from our necessary form for the mean of the reverse momentum refreshment. These works use $h\tilde{p}_{t+\delta}$ as the mean instead of $hf_\theta(t + \delta, x_t, \tilde{p}_{t+\delta})$. We again transition to discretized notation with $\delta := T/K$, and $k = 0, ..., K$. In this case, the log importance weight, corresponding to the log evidence estimate, satisfies

$$\log w_\theta(\mathbf{x}, \mathbf{p}) = \log \frac{\gamma(x_K)\mathcal{N}(p_K; 0, M)}{\pi_0(x_0)\mathcal{N}(p_0; 0, M)} + \sum_{k=1}^K \log \frac{\mathcal{N}(p_{k-1}; hf_\theta(k, x_{k-1}, \tilde{p}_k), (1 - h^2)M)}{\mathcal{N}(\tilde{p}_k; hp_{k-1}, (1 - h^2)M)}, \quad (21)$$

where $(\mathbf{x}, \mathbf{p})$ denote all the variables introduced by our integration scheme. We can show informally that minimizing $D_{\mathrm{KL}}(Q||P_\theta)$, i.e. maximizing the ELBO given $\mathbb{E}_Q[\log w_\theta(\mathbf{x}, \mathbf{p})]$, again corresponds to minimizing a score matching type loss (18) when $\delta \ll 1$; see Appendix C. Pseudo-code for our approach can be found in Algorithm 2.

## 4 Experiments

We run a number of experiments estimating normalizing constants to validate our approach, MCD and compare to differentiable AIS with ULA [53, 51] and Unadjusted Hamiltonian Annealing (UHA) [18, 54]. We first investigate the performance of these approaches on static target distributions using the same, fixed initial distribution and annealing schedule. Finally, we explore the performance of the methods for VAEs. Here, being the most expensive of our experiments, we include runtime comparisons of our method compared to baselines. Additional results on a Normalizing Flow target can be found in Appendix F.2. Full experimental details, chosen hyper-parameters, and model architectures can be found in Appendix E.

Our score model is parameterized by an MLP with residual connections that is conditioned on integration time $t$, and on the momentum term for the Hamiltonian case (see Algorithm 2). For an ablation on various network architectures we refer the reader to Appendix F.3.

---

[3][18] were not attempting to discretize an underdamped Langevin dynamics.

## 4.1 Static Targets

We estimate the normalizing constants of five simple distributions with known normalizing constants equal to $\log Z = 0$; $\mathcal{N}(10, I)$, $\mathcal{N}(0, 0.1I)$, a Gaussian mixture with 8 components whose means are drawn from $\mathcal{N}(3, I)$ where each component has variance 1, a standard Laplace distribution and a Student's T distribution with 3 degrees of freedom. We use $\mathcal{N}(0, I)$ as our initial distribution for all targets except for the Gaussian mixture and $\mathcal{N}(0, 0.1I)$ which both use $\mathcal{N}(0, 3^2I)$. We run each method using $K \in \{64, 256\}$ steps and use a fixed, linear annealing schedule. For all methods, sampling step-sizes per-timestep are tuned to via gradient descent to maximize the ELBO and the diagonal mass matrix is learned for the Hamiltonian samplers. Gaussian Mixture and Student-T results can be found in Tables 1 and 2, respectively. Additional results can be found in Appendix F.1.

| Sampler | ULA | | UHA | | ULA-MCD | | UHA-MCD | |
|---|---|---|---|---|---|---|---|---|
| # steps | 64 | 256 | 64 | 256 | 64 | 256 | 64 | 256 |
| Dim-20 | -0.47 $\pm$ 0.22 | -0.13 $\pm$ 0.07 | -0.03 $\pm$ 0.18 | 0.015 $\pm$ 0.03 | 0.01 $\pm$ 0.02 | 0.01 $\pm$ 0.01 | 0.01 $\pm$ 0.02 | -0.01 $\pm$ 0.01 |
| Dim-200 | -85.62 $\pm$ 2.01 | -21.98 $\pm$ 1.35 | -8.20 $\pm$ 1.84 | -1.26 $\pm$ 0.35 | -0.25 $\pm$ 0.03 | 0.08 $\pm$ 0.122 | 0.20 $\pm$ 0.49 | -0.05 $\pm$ 0.04 |
| Dim-500 | -304.06 $\pm$ 5.48 | -83.50 $\pm$ 7.88 | -44.45 $\pm$ 3.24 | -8.60 $\pm$ 1.69 | -2.61 $\pm$ 1.26 | 1.01 $\pm$ 0.99 | -1.74 $\pm$ 1.25 | -1.02 $\pm$ 0.03 |

Table 1: $\log Z$ estimates for a Gaussian mixture target. Averages and standard errors over 3 seeds.

On average and as expected, UHA outperforms ULA on most targets. Further our approximation to the optimal backward kernels can yield considerable improvements – enabling the ULA sampler to produce better results than UHA (using the standard AIS backward kernels). Additionally, we see that our approximation to the optimal backward kernels improves the performance of UHA as well.

We further emphasize that often ULA-MCD and UHA-MCD with 64 steps outperform or is on-par with ULA and UHA with 256 steps, a difference of factor 4 in terms of target gradient evaluations. As we show below in Table 3, the additional computational costs of fitting the score model of our method are only roughly twice that of the baselines.

| Sampler | ULA | | UHA | | ULA-MCD | | UHA-MCD | |
|---|---|---|---|---|---|---|---|---|
| # steps | 64 | 256 | 64 | 256 | 64 | 256 | 64 | 256 |
| Dim-20 | -0.09 $\pm$ 0.02 | -0.02 $\pm$ 0.01 | -0.04 $\pm$ 0.02 | -0.02 $\pm$ 0.01 | -0.06 $\pm$ 0.02 | -0.00 $\pm$ 0.01 | -0.03 $\pm$ 0.04 | -0.01 $\pm$ 0.01 |
| Dim-200 | -1.63 $\pm$ 0.23 | -0.88 $\pm$ 0.15 | -0.36 $\pm$ 0.34 | -0.10 $\pm$ 0.10 | -0.82 $\pm$ 0.28 | -0.30 $\pm$ 0.38 | -0.48 $\pm$ 0.20 | -0.10 $\pm$ 0.07 |
| Dim-500 | -5.43 $\pm$ 0.78 | -3.10 $\pm$ 0.07 | -2.86 $\pm$ 0.36 | -1.03 $\pm$ 0.13 | -4.00 $\pm$ 0.51 | -2.23 $\pm$ 0.18 | -2.03 $\pm$ 0.18 | 0.06 $\pm$ 0.30 |

Table 2: $\log Z$ estimates for a Student-T target. Averages and standard errors over 3 seeds.

## 4.2 Application to Amortized Inference

Next, we explore the application of our method to amortized inference, in the context of VAEs [31, 41]. These models are trained to infer latent representations using an inference neural network that consumes an input and produces parameters of an approximation to the true posterior distribution of the underlying generative model. In particular, this posterior distribution is different for each input. When applying AIS to VAE inference [51, 18, 54], the output of the inference network parameterizes the initial distribution of the annealing sequence to the true posterior. By training this end-to-end, we effectively learn the initial distribution for the diffusion process. Consequently, the diffusion marginals and their score vectors $\nabla \log q_t$ are different for every input, and we need to condition our score model $s_\theta$ on the inputs to reflect that. We achieve this by projecting the last hidden layer of the inference network into a summary vector that is concatenated to the other conditioning inputs of $s_\theta$.

We train a VAE on the binarized MNIST dataset [43], re-using architectures proposed in [18, 8] (two-layer MLP encoder/decoder, Bernoulli likelihood). All generative models use the same architecture

and hyper-parameters. We compare standard amortized variational inference with annealed ULA and UHA with standard AIS backward transition kernels, as well as ULA and UHA with our MCD transition kernels. We match the number of sampler steps between ULA/ULA-MCD and UHA/UHA-MCD to 64 and 32 respectively. ELBO and log-likelihood values on the test set are presented in Table 3.

| Sampler | VI | ULA | UHA | ULA-MCD | UHA-MCD |
|---|---|---|---|---|---|
| ELBO | $-96.32 \pm 0.40$ | $-90.41 \pm 0.17$ | $-88.58 \pm 0.51$ | $-90.10 \pm 0.10$ | $-88.08 \pm 0.07$ |
| Log-lik. | $-69.35 \pm 0.36$ | $-62.43 \pm 0.25$ | $-61.05 \pm 1.84$ | $-61.83 \pm 0.42$ | $-58.58 \pm 0.34$ |
| Iteration time | 0.024s | 0.055s | 0.050s | 0.101s | 0.098s |
| Total time | 3263.40s | 4694.52s | 5072.19s | 11304.36s | 11345.59s |

Table 3: Test set performance for MNIST VAE. Averages and standard errors over 5 seeds. We additionally report runtimes for a single training iteration, and total experiment time.

We see that, as reported in prior works [18, 51, 54], Monte Carlo based inference methods provide a significant benefit over standard amortized variational inference. In addition, our learned backward kernels lead to improved performance of both ULA and UHA, with the difference being more distinct for the latter. We also note that UHA has significantly larger error bars than all other methods, but UHA-MCD does not appear to inherit this. We further provide a runtime comparison in Table 3, both for a single training iteration, and total experiment time (including evaluation). We find that fitting the score models of our method results in roughly twice as much runtime (recall we fixed the number of sampler steps).

## 5   Limitations

While we have demonstrated that our optimized backward kernels can lead to large improvements over standard AIS backward kernels, our proposed approach has a number of limitations. First, since our method relies upon unadjusted Langevin or Hamiltonian sampling, we inherit many of the issues with these approaches. Using unadjusted samplers enables a fully-differentiable ELBO estimate which can in theory be used to tune the many parameters of the sampler. However, these samplers require repeated gradient steps as an inner-loop which can lead to divergence dynamics and numerical instability, making optimization difficult. Next, our proposed training procedure has a notable increase in memory consumption as we store the entire sampling trajectory in memory to train our neural network. We further note that it is possible to compute our importance weights online using $\mathcal{O}(1)$ memory. We could utilize this with Monte Carlo sub-sampling of the timestep (as with SGM [24]) to derive a constant memory training procedure for our approximate time-reversal at the cost of increased variance but we leave exploring this to future work. Finally, we observed that, without mitigation, parameterized forward samplers $Q_\phi$ sometimes dropped modes during training. Since this is a known challenge of using reverse KL, this might be improved by using alternative optimization objectives [35, 36].

## 6   Discussion

In this work we have explored AIS using unadjusted Langevin and Hamiltonian dynamics. We have demonstrated that the backward transition kernels typically used are suboptimal and we have presented the form of the optimal variance-minimizing backward kernels. We have further shown how an approximation to these kernels can be learned using score matching and that this objective corresponds to maximizing the ELBO in the limit of infinitesimally small time-discretization step-sizes. We have illustrated the benefit of using our proposed optimized backward kernels on a number of inference problems including fixed targets and amortized tasks with model learning.

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
