# A Variance calculations

We provide here, for a simple example, the variance expressions for $\log w_{\mathrm{ais}}(x_{0:K})$ and $\log w_{\mathrm{mar}}(x_K)$. All the expectations in this section are w.r.t. the proposal $Q$.

We consider the scenario where $\pi_0(x) = \gamma_0(x) = \mathcal{N}(x; 0, \sigma_0^2)$ and for $k \geq 1$

$$\pi_k(x) = \mathcal{N}(x; 0, \sigma_k^2), \quad \gamma_k(x) = \exp\left(-\frac{x^2}{2\sigma_k^2}\right).$$

We select the sequence of variances as follows

$$\sigma_k^2 = \left(\sigma_0^2\right)^{1-\frac{k}{K}} \left(\sigma^2\right)^{\frac{k}{K}} = \left(\frac{\sigma^2}{\sigma_0^2}\right)^{1/K} \sigma_{k-1}^2 := \beta_K \sigma_{k-1}^2$$

so that $\pi(x) = \pi_K(x) = \mathcal{N}(x; 0, \sigma^2)$. We will pick $\sigma^2 < \sigma_0^2$ so $\beta_K < 1$. Finally we consider the following proposal $Q$. At initialization $x_0 \sim \pi_0$ and for $k \geq 1$ we have the following transitions kernels

$$F_k(x'|x) = \mathcal{N}(x'; \alpha x, (1 - \alpha^2)\sigma_k^2)$$

which are $\pi_k$-invariant and $\alpha$ determines how fast it mixes. For $\alpha = 0$, we have exact samples from $\pi_k$ and as $\alpha \to 1$ we are mixing less and less. For this setup, it is possible to provide exact calculations for the expectation and variance of the log-evidence estimate.

**Proposition 4.** *Under $Q$, we have $x_k \sim \mathcal{N}(0, \xi_k^2)$ where $\xi_0^2 = \sigma_0^2$ and*

$$\xi_k^2 = \alpha^2 \xi_{k-1}^2 + (1 - \alpha^2)\sigma_k^2.$$

*The expectation of the log evidence estimates satisfies*

$$\mathbb{E}[\log w_{\mathrm{mar}}(x_K)] = \frac{1}{2}\log\left(2\pi\xi_K^2\right) + \frac{1}{2}\left(\frac{1}{\xi_K^2} - \frac{1}{\sigma_K^2}\right)\xi_K^2,$$

$$\mathbb{E}[\log w_{\mathrm{ais}}(x_{0:K})] = \frac{1}{2}\log(2\pi\sigma_0^2) + \frac{(\beta_K - 1)}{2}\sum_{k=1}^{K}\frac{\xi_{k-1}^2}{\sigma_k^2},$$

*while their variance is given by*

$$\mathrm{var}[\log w_{\mathrm{mar}}(x_K)] = \frac{1}{2}\left(\frac{1}{\xi_K^2} - \frac{1}{\sigma_K^2}\right)^2\xi_K^4,$$

$$\mathrm{var}[\log w_{\mathrm{ais}}(x_{0:K})] = (\beta_K - 1)^2\left(\sum_{k=1}^{K}\frac{\xi_{k-1}^4}{2\sigma_k^4} + \sum_{K \geq l > k \geq 1}\frac{\alpha^{2(l-k)}\xi_{k-1}^4}{\sigma_k^2\sigma_l^2}\right).$$

*Proof.* The proposal is given by $x_0 \sim \mathcal{N}(0, \sigma_0^2)$ and

$$x_k = \alpha x_{k-1} + \sqrt{1 - \alpha^2}\sigma_k\epsilon_k, \qquad \epsilon_k \overset{\text{i.i.d.}}{\sim} \mathcal{N}(0, 1).$$

So marginally, under $Q$, we have $x_k \sim \mathcal{N}(0, \xi_k^2)$ where $(\xi_k)_{k \geq 0}$ satisfies the recursion stated above.

We are first looking at the optimal log-estimate of $Z$ which is given by

$$\log w_{\mathrm{mar}}(x_K) = \log \gamma_K(x_K) - \log q_K(x_K)$$
$$= \frac{1}{2}\log\left(2\pi\xi_K^2\right) + \frac{1}{2}\left(\frac{1}{\xi_K^2} - \frac{1}{\sigma_K^2}\right)x_K^2$$

so we have

$$\mathbb{E}[\log w_{\mathrm{mar}}(x_K)] = \frac{1}{2}\log\left(2\pi\xi_K^2\right) + \frac{1}{2}\left(\frac{1}{\xi_K^2} - \frac{1}{\sigma_K^2}\right)\xi_K^2$$

and, using $\mathrm{var}[x^2] = 2\sigma^4$ for $x \sim \mathcal{N}(0, \sigma^2)$, we obtain

$$\mathrm{var}[\log w_{\mathrm{mar}}(x_K)] = \frac{1}{2}\left(\frac{1}{\xi_K^2} - \frac{1}{\sigma_K^2}\right)^2\xi_K^4.$$

We are now looking at the AIS log-estimate of $Z$ which is given by

$$\log w_{\text{ais}}(x_{0:K}) = \sum_{k=1}^{K} \log(\gamma_k(x_{k-1})) - \log(\gamma_{k-1}(x_{k-1}))$$

$$= \frac{1}{2} \log(2\pi\sigma_0^2) + \frac{(\beta_K - 1)}{2} \sum_{k=1}^{K} \frac{x_{k-1}^2}{\sigma_k^2},$$

the term $\frac{1}{2} \log(2\pi\sigma_0^2)$ coming from the fact that we consider $\gamma_0 = \pi_0$. It follows that

$$\mathbb{E}[\log w_{\text{ais}}(x_{0:K})] = \frac{1}{2} \log(2\pi\sigma_0^2) + \frac{(\beta_K - 1)}{2} \sum_{k=1}^{K} \frac{\xi_{k-1}^2}{\sigma_k^2}.$$

The variance is given by

$$\text{var}[\log w_{\text{ais}}(x_{0:K})] = \frac{(\beta_K - 1)^2}{4} \text{var} \left[ \sum_{k=1}^{K} \frac{x_{k-1}^2}{\sigma_k^2} \right]$$

$$= \frac{(\beta_K - 1)^2}{4} \sum_{k=1}^{K} \frac{1}{\sigma_k^4} \text{var} \left[ x_{k-1}^2 \right] + 2 \frac{(\beta_K - 1)^2}{4} \sum_{K \geq l > k \geq 1} \frac{1}{\sigma_k^2 \sigma_l^2} \text{cov} \left[ x_{k-1}^2, x_{l-1}^2 \right].$$

Now, using again $\text{var}[x^2] = 2\sigma^4$ for $x \sim \mathcal{N}(0, \sigma^2)$, we have $\text{var}\left[x_{k-1}^2\right] = 2\xi_{k-1}^4$. To compute $\text{cov}\left[x_{k-1}^2, x_{l-1}^2\right]$, we use the fact that one can easily check from the form of the forward transitions that $(x_{k-1}, x_{l-1})$ satisfies for $l > k$

$$\text{cov}(x_{k-1}, x_{l-1}) = \alpha^{l-k}\xi_{k-1}^2.$$

So we have in distribution for $z \sim \mathcal{N}(0, 1)$ independent of $x_{k-1}$

$$(x_{k-1}, x_{l-1}) = (x_{k-1}, \alpha^{l-k}x_{k-1} + \sqrt{\xi_{l-1}^2 - \alpha^{2(l-k)}\xi_{k-1}^2} z).$$

Thus it follows that

$$x_{k-1}^2 x_{l-1}^2 = x_{k-1}^2 \left( \alpha^{l-k}x_{k-1} + \sqrt{\xi_{l-1}^2 - \alpha^{2(l-k)}\xi_{k-1}^2} z \right)^2$$

$$= x_{k-1}^2 \left( \alpha^{2(l-k)}x_{k-1}^2 + (\xi_{l-1}^2 - \alpha^{2(l-k)}\xi_{k-1}^2)z^2 + 2\alpha^{l-k}\sqrt{\xi_{l-1}^2 - \alpha^{2(l-k)}\xi_{k-1}^2} x_{k-1}z \right)$$

$$= \alpha^{2(l-k)}x_{k-1}^4 + (\xi_{l-1}^2 - \alpha^{2(l-k)}\xi_{k-1}^2)z^2 x_{k-1}^2 + 2\alpha^{l-k}\sqrt{\xi_{l-1}^2 - \alpha^{2(l-k)}\xi_{k-1}^2} x_{k-1}^3 z.$$

Hence, we obtain

$$\text{cov}\left[x_{k-1}^2, x_{l-1}^2\right] = \mathbb{E}\left[x_{k-1}^2 x_{l-1}^2\right] - \mathbb{E}\left[x_{k-1}^2\right] \mathbb{E}\left[x_{l-1}^2\right]$$

$$= 3\alpha^{2(l-k)}\xi_{k-1}^4 + (\xi_{l-1}^2 - \alpha^{2(l-k)}\xi_{k-1}^2)\xi_{k-1}^2 - \xi_{k-1}^2\xi_{l-1}^2$$

$$= 2\alpha^{2(l-k)}\xi_{k-1}^4.$$

This finally yields

$$\text{var}[\log w_{\text{ais}}(x_{0:K})] = \frac{(\beta_K - 1)^2}{4} \sum_{k=1}^{K} \frac{1}{\sigma_k^4} \text{var} \left[ x_{k-1}^2 \right] + 2 \frac{(\beta_K - 1)^2}{4} \sum_{K \geq l > k \geq 1} \frac{1}{\sigma_k^2 \sigma_l^2} \text{cov} \left[ x_{k-1}^2, x_{l-1}^2 \right]$$

$$= \frac{(\beta_K - 1)^2}{2} \sum_{k=1}^{K} \frac{\xi_{k-1}^4}{\sigma_k^4} + (\beta_K - 1)^2 \sum_{K \geq l > k \geq 1} \frac{\alpha^{2(l-k)}\xi_{k-1}^4}{\sigma_k^2 \sigma_l^2},$$

as required. $\qquad\square$

We now illustrate these results in Figure 3 by plotting the expectation and the root mean square error of $\log w_{\text{mar}}$ and $\log w_{\text{ais}}$ for various $\alpha$ and $K$.

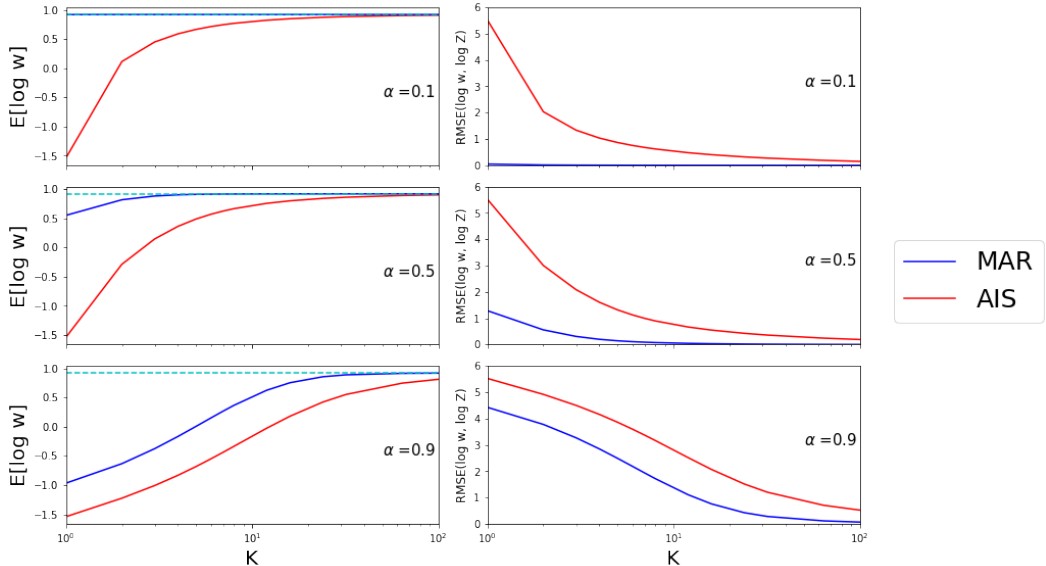

Figure 3: Plot of analytic results from Section A. The left column shows the analytic mean $\mathbb{E}[\log w]$ as a function of the number of temperature transitions $K$ for different values of the mixing parameter $\alpha \in [0, 1]$. $\alpha = 0$ corresponds to perfect mixing while $\alpha = 1$ corresponds to no mixing. MAR denotes the optimal importance weight, where AIS shows the one from Annealed Importance Sampling. Both estimators tend to the correct value $\log Z$ (shown as a Cyan line) as $K$ becomes large but MAR does so faster. The right column shows the same plots but for the root mean squared error (RMSE) of $\log w$ treated as an estimator of $\log Z$. It is computed as the root of the bias squared plus the variance, i.e $\sqrt{(\mathbb{E}[\log w] - \log Z)^2 + \text{Var}[\log w]}$. The RMSE of both estimators tends to zero in both cases as $K$ becomes large but MAR does so faster.

## B    Proof of Propositions

### B.1    Proof of Proposition 1

*Proof.* The chain rule for the Kullback–Leibler divergence $D_{\mathrm{KL}}(Q||P)$ yields

$$D_{\mathrm{KL}}(Q||P) = D_{\mathrm{KL}}(q_K||\pi) + \mathbb{E}_{q_K}\Big[D_{\mathrm{KL}}(Q(\cdot|x_K)||P(\cdot|x_K))\Big], \tag{22}$$

where, from (2) and (3), the conditional distributions of $x_{0:K-1}$ given $x_K$ are equal to

$$Q(x_{0:K-1}|x_K) = \prod_{k=0}^{K-1} B_k^{\mathrm{opt}}(x_k|x_{k+1}), \qquad P(x_{0:K-1}|x_K) = \prod_{k=0}^{K-1} B_k(x_k|x_{k+1}), \tag{23}$$

The expression for $Q$ above follows directly from its time-reversed decomposition; i.e.

$$Q(x_{0:K}) = q_K(x_K) \prod_{k=0}^{K-1} \frac{q_k(x_k)F_{k+1}(x_{k+1}|x_k)}{q_{k+1}(x_{k+1})} = q_K(x_K) \prod_{k=0}^{K-1} B_k^{\mathrm{opt}}(x_k|x_{k+1}), \tag{24}$$

where we recall that $q_0(x_0) = \pi_0(x_0)$. It thus follows directly from (22) and (23) that the backward transition kernels $(B_k)_{k=0}^{K-1}$ minimizing $D_{\mathrm{KL}}(Q||P)$ are $(B_k^{\mathrm{opt}})_{k=0}^{K-1}$ as this implies $P(x_{0:K-1}|x_K) = Q(x_{0:K-1}|x_K)$.

The variance decomposition formula yields for all $P$

$$\begin{aligned}
\text{var}_Q[w(x_{0:K})] &= \text{var}_{q_K}[\mathbb{E}_{Q(\cdot|x_K)}[w(x_{0:K})]] + \mathbb{E}_{q_K}[\text{var}_{Q(\cdot|x_K)}[w(x_{0:K})]] \\
&= \text{var}_{q_K}[w_{\mathrm{mar}}(x_K)] + \mathbb{E}_{q_K}[\text{var}_{Q(\cdot|x_K)}[w(x_{0:K})]] \\
&\geq \text{var}_{q_K}[w_{\mathrm{mar}}(x_K)].
\end{aligned}$$

By direct calculations, we also have $w_{\mathrm{mar}}(x_K) = \Gamma^{\mathrm{opt}}(x_{0:K})/Q(x_{0:K})$ so $P^{\mathrm{opt}}$ minimizes the variance of the evidence estimate. $\qquad\square$

## B.2 Proof of Proposition 2

We establish first here Proposition 5 and Proposition 6. Both results can then be easily combined to obtain Proposition 2.

**Proposition 5.** *Under regularity conditions, one has*

$$D_{\mathrm{KL}}(\mathcal{Q}||\mathcal{P}_\theta) = \mathbb{E}_{\mathcal{Q}}\Big[ \int_0^T ||s_\theta(t, x_t) - \nabla \log q_t(x_t)||^2 \mathrm{d}t\Big] + C_1 \tag{25}$$

$$= \sum_{k=1}^{K} \int_{t_{k-1}}^{t_k} \mathbb{E}_{\mathcal{Q}}\left[||s_\theta(t, x_t) - \nabla \log q_{t|t_{k-1}}(x_t|x_{t_{k-1}})||^2\right] \mathrm{d}t + C_2, \tag{26}$$

*for constants $C_1, C_2$ independent of $\theta$, where $t_k = k\delta$ with $K = T/\delta$ and $q_{t|s}(x'|x)$ is the density of $X_t = x'$ given $X_s = x$ under $\mathcal{Q}$.*

To establish (25), we follow arguments similar to [47, Theorem 2]. The loss (26) we then consider differs from the one uses in the score-based generative modeling literature. This is because, contrary to the Ornstein–Ulhenbeck process used for SGM, the transition density $q_{t'|t}(x'|x)$ of the forward diffusion (10) is not available in closed-form and can only be approximated reliably when $t' - t$ is small. Practically, to obtain a tractable criterion, we need to first approximate the integrals in (26) by the rectangular rule. We also discretize the Langevin dynamics using an Euler–Maruyama scheme; i.e. we use an approximation $Q$ of $\mathcal{Q}$ based on the ULA kernel $F_k(x_k|x_{k-1}) = \mathcal{N}(x_k; x_{k-1} + \delta\nabla \log \pi_k(x_{k-1}); 2\delta I)$ approximating $q_{t_k|t_{k-1}}(x'|x)$. We recall here that we slightly abuse notation by writing $\pi_{t_k} = \pi_{k\delta}$ as $\pi_k$. We thus finally obtain a loss

$$\mathcal{L}(\theta) = \delta \sum_{k=1}^{K} \mathbb{E}_Q\left[||s_\theta(k, x_k) - \nabla \log F_k(x_k|x_{k-1})||^2\right]. \tag{27}$$

*Proof.* We assume here sufficient regularity conditions ensuring that the SDEs given below admit a unique solution, their corresponding time-reversed diffusions are well-defined and Girsanov theorem applies; see e.g. [47, Appendix A].

By the chain rule for KL divergence, one has

$$D_{\mathrm{KL}}(\mathcal{Q}||\mathcal{P}_\theta) = D_{\mathrm{KL}}(q_T||\pi) + \mathbb{E}_{q_T}\Big[D_{\mathrm{KL}}(\mathcal{Q}(\cdot|x_T)||\mathcal{P}_\theta(\cdot|x_T))\Big] \tag{28}$$

where $\mathcal{Q}(\cdot|x_T)$ and $\mathcal{P}_\theta(\cdot|x_T)$ are the path measures induced by

$$\mathrm{d}\bar{x}_t = \big\{ - \nabla \log \pi_{T-t}(\bar{x}_t) + 2\nabla \log q_{T-t}(\bar{x}_t)\big\}\mathrm{d}t + \sqrt{2}\mathrm{d}\bar{B}_t, \qquad \bar{x}_0 = x_T, \tag{29}$$

and

$$\mathrm{d}\bar{x}_t = \big\{ - \nabla \log \pi_{T-t}(\bar{x}_t) + 2\nabla \log s_\theta(T - t, \bar{x}_t)\big\}\mathrm{d}t + \sqrt{2}\mathrm{d}\bar{B}_t, \qquad \bar{x}_0 = x_T. \tag{30}$$

We now use Girsanov theorem (see e.g. [32, Section 10.3] and [39]) to compute the Radon–Nikodym derivative $\mathrm{d}\mathcal{Q}(\cdot|x_T)/\mathrm{d}\mathcal{P}_\theta(\cdot|x_T)$ so that

$$\mathbb{E}_{q_T}\Big[D_{\mathrm{KL}}(\mathcal{Q}(\cdot|x_T)||\mathcal{P}_\theta(\cdot|x_T))\Big]$$

$$= - \mathbb{E}_{\mathcal{Q}}\Big[ \log \frac{\mathrm{d}\mathcal{P}_\theta(\cdot|x_T)}{\mathrm{d}\mathcal{Q}(\cdot|x_T)}\Big]$$

$$= \mathbb{E}_{\mathcal{Q}}\Big[\sqrt{2} \int_0^T (\nabla \log q_{T-t}(\bar{x}_t) - s_\theta(T - t, \bar{x}_t))\mathrm{d}\bar{B}_t + \int_0^T ||\nabla \log q_{T-t}(\bar{x}_t) - s_\theta(T - t, \bar{x}_t)||^2 \mathrm{d}t\Big]$$

$$= \mathbb{E}_{\mathcal{Q}}\Big[ \int_0^T ||\nabla \log q_t(x_t) - s_\theta(t, x_t)||^2 \mathrm{d}t\Big],$$

as $\mathbb{E}_{\mathcal{Q}}\Big[ \int_0^T f_t(\bar{x}_t)\mathrm{d}\bar{B}_t\Big] = 0$ for any function $f_t$.

As in [11] in a different context, we can write for any partition of $[0, T]$ defined by $t_0 = 0 < t_1 < \cdots < t_{K-1} < t_K = T$

$$\mathbb{E}_{\mathcal{Q}}\Big[\int_0^T ||\nabla \log q_t(x_t) - s_\theta(t, x_t)||^2 \mathrm{d}t\Big] = \int_0^T \int ||\nabla \log q_t(x) - s_\theta(t, x)||^2 q_t(x) \mathrm{d}x \mathrm{d}t$$

$$= \sum_{k=1}^K \int_{t_{k-1}}^{t_k} \int ||\nabla \log q_t(x) - s_\theta(t, x)||^2 q_t(x) \mathrm{d}x \mathrm{d}t$$

where, for a constant $c$ independent of $\theta$, we have

$$\int_{t_{k-1}}^{t_k} \int ||\nabla \log q_t(x) - s_\theta(t, x)||^2 q_t(x) \mathrm{d}x \mathrm{d}t$$

$$= \int_{t_{k-1}}^{t_k} \int \left\{ ||\nabla \log q_t(x)||^2 + ||s_\theta(t, x)||^2 - 2 s_\theta(t, x)^{\mathrm{T}} \nabla \log q_t(x) \right\} q_t(x) \mathrm{d}x \mathrm{d}t$$

$$= \int_{t_{k-1}}^{t_k} \int \left\{ ||s_\theta(t, x)||^2 - 2 s_\theta(t, x)^{\mathrm{T}} \nabla \log q_t(x) \right\} q_t(x) \mathrm{d}x \mathrm{d}t + c.$$

Now we have

$$\int_{t_{k-1}}^{t_k} \int s_\theta(t, x)^{\mathrm{T}} \nabla \log q_t(x) q_t(x) \mathrm{d}x \mathrm{d}t = \int_{t_{k-1}}^{t_k} \int s_\theta(t, x)^{\mathrm{T}} \nabla q_t(x) \mathrm{d}x \mathrm{d}t \qquad (31)$$

where, using Chapman-Kolmogorov, $q_t$ satisfies

$$q_t(x) = \int q_{t_{k-1}}(x_{t_{k-1}}) q_{t|t_{k-1}}(x|x_{t_{k-1}}) \mathrm{d}x_{t_{k-1}}. \qquad (32)$$

It follows that

$$\nabla q_t(x) = \int q_{t_{k-1}}(x_{t_{k-1}}) \nabla q_{t|t_{k-1}}(x|x_{t_{k-1}}) \mathrm{d}x_{t_{k-1}}. \qquad (33)$$

Hence, we have

$$\int_{t_{k-1}}^{t_k} \int s_\theta(t, x)^{\mathrm{T}} \nabla q_t(x) \mathrm{d}x \mathrm{d}t$$

$$= \int_{t_{k-1}}^{t_k} \int \int s_\theta(t, x)^{\mathrm{T}} \nabla \log q_{t|t_{k-1}}(x|x_{t_{k-1}}) q_{t_{k-1}}(x_{t_{k-1}}) q_{t|t_{k-1}}(x|x_{t_{k-1}}) \mathrm{d}x_{t_{k-1}} \mathrm{d}x \mathrm{d}t$$

so minimizing $\mathbb{E}_{q_T}\Big[ D_{\mathrm{KL}}(\mathcal{Q}(\cdot|x_T)||\mathcal{P}_\theta(\cdot|x_T)) \Big]$ w.r.t. $\theta$ is equivalent to minimize

$$\sum_{k=1}^K \int_{t_{k-1}}^{t_k} \int \int ||s_\theta(t, x)||^2 q_{t_{k-1}}(x_{t_{k-1}}) q_{t|t_{k-1}}(x|x_{t_{k-1}}) \mathrm{d}x_{t_{k-1}} \mathrm{d}x \mathrm{d}t$$

$$-2 \sum_{k=1}^K \int_{t_{k-1}}^{t_k} \int \int s_\theta(t, x)^{\mathrm{T}} \nabla \log q_{t|t_{k-1}}(x|x_{t_{k-1}}) q_{t_{k-1}}(x_{t_{k-1}}) q_{t|t_{k-1}}(x|x_{t_{k-1}}) \mathrm{d}x_{t_{k-1}} \mathrm{d}x \mathrm{d}t$$

$$= \sum_{k=1}^K \int_{t_{k-1}}^{t_k} \int \int ||s_\theta(t, x) - \nabla \log q_{t|t_{k-1}}(x|x_{t_{k-1}})||^2 q_{t_{k-1}}(x_{t_{k-1}}) q_{t|t_{k-1}}(x|x_{t_{k-1}}) \mathrm{d}x_{t_{k-1}} \mathrm{d}x \mathrm{d}t + C$$

where $C$ is independent of $\theta$. Hence, this is equivalent to minimizing (26). $\qquad \square$

We now establish results about the discrete-time Kullback–Leibler divergence $D_{\mathrm{KL}}(Q||P_\theta)$. First note that

$$
\begin{aligned}
D_{\mathrm{KL}}(Q||P_\theta) &= \mathbb{E}_Q\left[\log\frac{Q(x_{0:K})}{P_\theta(x_{0:K})}\right] \\
&= \mathbb{E}_Q\left[\log\frac{\pi_0(x_0)\prod_{k=0}^{K-1}F_{k+1}(x_{k+1}|x_k)}{\pi(x_K)\prod_{k=0}^{K-1}B_k^\theta(x_k|x_{k+1})}\right] \\
&= -\sum_{k=0}^{K-1}\mathbb{E}_Q\left[\log B_k^\theta(x_k|x_{k+1})\right] + C_1,
\end{aligned} \tag{34}
$$

where, as $B_k^\theta(x_k|x_{k+1}) = \mathcal{N}(x_k; x_{k+1} - \delta\nabla\log\pi_{k+1}(x_{k+1}) + 2\delta s_\theta(k+1, x_{k+1}), 2\delta I)$, one has

$$
\begin{aligned}
-\log B_k^\theta(x_k|x_{k+1}) &= \frac{1}{4\delta}||x_k - x_{k+1} + \delta\nabla\log\pi_{k+1}(x_{k+1}) - 2\delta s_\theta(k+1, x_{k+1})||^2 + C_2 \\
&= \delta\left\|s_\theta(k+1, x_{k+1}) - \frac{1}{2\delta}(x_k - x_{k+1} + \delta\nabla\log\pi_{k+1}(x_{k+1}))\right\|^2 + C_2 \tag{35} \\
&\approx \delta\left\|s_\theta(k+1, x_{k+1}) - \frac{1}{2\delta}(x_k - x_{k+1} + \delta\nabla\log\pi_{k+1}(x_k))\right\|^2 + C_2 \\
&= \delta\left\|s_\theta(k+1, x_{k+1}) - \nabla\log F_{k+1}(x_{k+1}|x_k)\right\|^2 + C_2, \tag{36}
\end{aligned}
$$

where we have used $\pi_{k+1}(x_{k+1}) \approx \pi_{k+1}(x_k)$ for $\delta \ll 1$. The sum over $k = 0, ..., K-1$ of the first terms on the r.h.s. of (35) are equal to the loss $\mathcal{L}(\theta)$ defined in (27). More rigorously, we can prove the following result.

**Assumption 1.** *There exists $L \leq \infty$ such that for all $k$ and $x, x' \in \mathbb{R}^d$*

$$
\left\|\nabla\log\pi_k(x) - \nabla\log\pi_k(x')\right\| \leq L\left\|x - x'\right\|. \tag{37}
$$

**Assumption 2.** *There exists $C \leq \infty$ such that*

$$
\limsup_K \max_{k=0,...,K-1} \mathbb{E}_{Q_K}\left[\left\|\nabla\log\pi_{k+1}(x_k)\right\|^2\right] \leq C \tag{38}
$$

*and for any $\theta$*

$$
\limsup_K \max_{k=0,...,K-1} \mathbb{E}_{Q_K}\left[\left\|\nabla_\theta s_\theta(k+1, x_{k+1})\right\|^2\right] \leq C, \tag{39}
$$

*where we have emphasized here notationally that $Q$ is a function of $K$.*

**Proposition 6.** *Under Assumptions 1-2, the gradient of the Kullback–Leibler divergence $D_{\mathrm{KL}}(Q||P_\theta)$ satisfies*

$$
\nabla D_{\mathrm{KL}}(Q||P_\theta) = \nabla\mathcal{L}(\theta) + \epsilon(\theta), \tag{40}
$$

*for $\mathcal{L}(\theta)$ defined in (27) and a function $\epsilon$ satisfying $\lim_{K\to\infty}\epsilon(\theta) = 0$.*

*Proof.* In the rest of the proof, all the expectations are taken w.r.t. $Q$ unless mentioned otherwise and we drop it from the notations for simplicity. However as we take gradients w.r.t. to both $x$ and $\theta$, this is indicated notationally to avoid confusion. We also assume that $\theta$ is a scalar in the proof, the extension to the multivariate case is straightforward.

Using (34), we have

$$
\nabla_\theta D_{\mathrm{KL}}(Q||P_\theta) = -\sum_{k=0}^{K-1}\mathbb{E}\left[\nabla_\theta\log B_k^\theta(x_k|x_{k+1})\right], \tag{41}
$$

where, from (35), one has

$$
\begin{aligned}
&-\nabla_\theta\log B_k^\theta(x_k|x_{k+1}) \tag{42} \\
&= \delta\nabla_\theta\left\|s_\theta(k+1, x_{k+1}) - \frac{1}{2\delta}(x_k - x_{k+1} + \delta\nabla_x\log\pi_{k+1}(x_{k+1}))\right\|^2 \\
&= 2\delta\nabla_\theta s_\theta(k+1, x_{k+1})^{\mathrm{T}}(s_\theta(k+1, x_{k+1}) - \frac{1}{2\delta}(x_k - x_{k+1} + \delta\nabla_x\log\pi_{k+1}(x_{k+1}))). \tag{43}
\end{aligned}
$$

We also have

$$\nabla_\theta \mathcal{L}(\theta) = \delta \sum_{k=0}^{K-1} \mathbb{E}\left[\nabla_\theta \left\|s_\theta(k, x_k) - \nabla_x \log F_k(x_k|x_{k-1})\right\|^2\right], \tag{44}$$

where

$$\begin{aligned}
&\delta \nabla_\theta \left\|s_\theta(k, x_k) - \nabla_x \log F_k(x_k|x_{k-1})\right\|^2 \\
=&\delta \nabla_\theta \left\|s_\theta(k, x_k) - \frac{1}{2\delta}(x_k - x_{k+1} + \delta \nabla_x \log \pi_{k+1}(x_k))\right\|^2 \\
=&2\delta \nabla_\theta s_\theta(k+1, x_{k+1})^{\mathrm{T}}(s_\theta(k+1, x_{k+1}) - \frac{1}{2\delta}(x_k - x_{k+1} + \delta \nabla_x \log \pi_{k+1}(x_k))). \tag{45}
\end{aligned}$$

So we obtain by using (34) and (35)

$$\nabla_\theta D_{\mathrm{KL}}(Q||P_\theta) = \nabla_\theta \mathcal{L}(\theta) + \epsilon(\theta), \tag{46}$$

for

$$\epsilon(\theta) = 2\delta \mathbb{E}\left[\sum_{k=0}^{K-1} \nabla_\theta s_\theta(k+1, x_{k+1})^{\mathrm{T}}(\nabla_x \log \pi_{k+1}(x_k) - \nabla_x \log \pi_{k+1}(x_{k+1}))\right]. \tag{47}$$

Hence we have

$$\begin{aligned}
|\epsilon(\theta)| &\leq 2\delta \sum_{k=0}^{K-1} \mathbb{E}\left[|\nabla_\theta s_\theta(k+1, x_{k+1})^{\mathrm{T}}(\nabla_x \log \pi_{k+1}(x_k) - \nabla_x \log \pi_{k+1}(x_{k+1}))|\right] \\
&\leq 2\delta \sum_{k=0}^{K-1} \mathbb{E}\left[\left\|\nabla_\theta s_\theta(k+1, x_{k+1})\right\|^2\right]^{1/2} \mathbb{E}\left[\left\|\nabla_x \log \pi_{k+1}(x_k) - \nabla_x \log \pi_{k+1}(x_{k+1})\right\|^2\right]^{1/2}
\end{aligned}$$
$$\tag{48}$$

From Assumption 1, we have

$$\begin{aligned}
\mathbb{E}\left[\left\|\nabla_x \log \pi_{k+1}(x_k) - \nabla_x \log \pi_{k+1}(x_{k+1})\right\|^2\right] &\leq L^2 \mathbb{E}\left[\left\|x_{k+1} - x_k\right\|^2\right] \\
&\leq 2L^2 \delta \mathbb{E}\left[\delta \left\|\nabla_x \log \pi_{k+1}(x_k)\right\|^2 + 2M\right], \tag{49}
\end{aligned}$$

where $M = \mathbb{E}_{Z \sim \mathcal{N}(0,I)}[||Z||^2]$ as $x_{k+1} = x_k + \delta \nabla_x \log \pi_{k+1}(x_k) + \sqrt{2\delta}Z$ under $Q$. Now using Assumption 2, it follows from (48), (49), (38), (39) and $K = O(1/\delta)$ that $\epsilon(\theta) = O(\sqrt{\delta})$. The result follows.

$\square$

## B.3 Proof of Proposition 3

We also assume here sufficient regularity conditions ensuring that the SDEs given below admit a unique solution, their corresponding time-reversed diffusions are well-defined and Girsanov theorem applies. The proof is very similar to the proof of the first part of Proposition 2 (i.e. Proposition 5 in Appendix B.2).

To start with, we use again the chain rule for KL divergences, one has

$$D_{\mathrm{KL}}(\mathcal{Q}||\mathcal{P}_\theta) = D_{\mathrm{KL}}(\eta_T||\pi_T \otimes \mathcal{N}(0, M)) + \mathbb{E}_{\eta_T}\left[D_{\mathrm{KL}}(\mathcal{Q}(\cdot|x_T, p_T)||\mathcal{P}_\theta(\cdot|x_T, p_T))\right] \tag{50}$$

where $\mathcal{Q}(\cdot|x_T, p_T)$ is the path measure induced by

$$\begin{aligned}
\mathrm{d}\bar{x}_t &= -M^{-1}\bar{p}_t \mathrm{d}t, \tag{51} \\
\mathrm{d}\bar{p}_t &= -\nabla \log \pi_{T-t}(\bar{x}_t)\mathrm{d}t + \zeta\bar{p}_t\mathrm{d}t + 2\zeta M \nabla_{\bar{p}_t} \log \eta_{T-t}(\bar{x}_t, \bar{p}_t)\mathrm{d}t + \sqrt{2\zeta}M^{1/2}\mathrm{d}\bar{B}_t
\end{aligned}$$

and $\mathcal{P}_\theta(\cdot|x_T, p_T)$ the path measure induced by

$$d\bar{x}_t = -M^{-1}\bar{p}_t dt, \tag{52}$$
$$d\bar{p}_t = -\nabla \log \pi_{T-t}(\bar{x}_t)dt + \zeta\bar{p}_t dt + 2\zeta M s_\theta(T-t, \bar{x}_t, \bar{p}_t) + \sqrt{2\zeta}M^{1/2}d\bar{B}_t,$$

these two diffusions being initialized at $(\bar{x}_0, \bar{p}_0) = (x_T, p_T)$.

By now applying a version of Girsanov's theorem that allows for some of the components of the diffusions to be noiseless [49, Theorem 4], we obtain

$$\mathbb{E}_{\eta_T}\Big[D_{\mathrm{KL}}(\mathcal{Q}(\cdot|x_T, p_T)||\mathcal{P}_\theta(\cdot|x_T, p_T))\Big]$$
$$=\frac{1}{2}\mathbb{E}_{\mathcal{Q}}\Big[\int_0^T \int ||2\zeta M\nabla_p \log \eta_t(x_t, p_t) - 2\zeta M s_\theta(t, x_t, p_t)||^2_{(2\zeta M)^{-1}}dt\Big]$$
$$=\zeta\mathbb{E}_{\mathcal{Q}}\Big[\int_0^T ||\nabla_p \log \eta_t(x_t, p_t) - s_\theta(t, x_t, p_t)||^2_M dt\Big]. \tag{53}$$

Now equation (18) follows directly from (50) and (53). We are now going to rewrite this loss to make it more tractable. We have for any partition of $[0, T]$ defined by $t_0 = 0 < t_1 < \cdots < t_{K-1} < t_K = T$

$$\mathbb{E}_{\mathcal{Q}}\Big[\int_0^T ||\nabla_p \log \eta_t(x_t, p_t) - s_\theta(t, x_t, p_t)||^2_M dt\Big]$$
$$=\int_0^T \int ||\nabla_p \log \eta_t(x, p) - s_\theta(t, x, p)||^2_M \, \eta_t(x, p)dxdpdt$$
$$=\sum_{k=1}^K \int_{t_{k-1}}^{t_k} \int ||\nabla_p \log \eta_t(x, p) - s_\theta(t, x, p)||^2_M \, \eta_t(x, p)dxdpdt$$

where

$$\int_{t_{k-1}}^{t_k} \int ||\nabla_p \log \eta_t(x, p) - s_\theta(t, x, p)||^2_M \eta_t(x, p)dxdpdt$$
$$=\int_{t_{k-1}}^{t_k} \int \big\{||\nabla_p \log \eta_t(x, p)||^2 + ||s_\theta(t, x, p||^2_M - 2s_\theta(t, x, p)^{\mathrm{T}}M\nabla_p \log \eta_t(x, p)\big\}\, \eta_t(x, p)dxdpdt$$
$$=\int_{t_{k-1}}^{t_k} \int \big\{||s_\theta(t, x, p)||^2_M - 2s_\theta(t, x, p)^{\mathrm{T}}M\nabla_p \log \eta_t(x, p)\big\}\, \eta_t(x, p)dxdpdt + c.$$

Now we have

$$\int_{t_{k-1}}^{t_k} \int s_\theta(t, x, p)^{\mathrm{T}}M\nabla_p \log \eta_t(x, p)\eta_t(x, p)dxdpdt = \int_{t_{k-1}}^{t_k} \int s_\theta(t, x, p)^{\mathrm{T}}M\nabla_p \eta_t(x, p)dxdpdt$$

where, using Chapman-Kolmogorov, we have

$$\eta_t(x, p) = \int \eta_{t_{k-1}}(x_{t_{k-1}}, p_{t_{k-1}})\eta_{t|t_{k-1}}(x, p|x_{t_{k-1}}, p_{t_{k-1}})dx_{t_{k-1}}dp_{t_{k-1}}.$$

Here $\eta_{t|t_{k-1}}(x, p|x_{t_{k-1}}, p_{t_{k-1}})$ denote the transition density of $(x_t, p_t) = (x, p)$ given $(x_{t_{k-1}}, p_{t_{k-1}})$ under the forward dynamics (15) so that

$$\nabla_p \eta_t(x, p) = \int \eta_{t_{k-1}}(x_{t_{k-1}}, p_{t_{k-1}})\nabla_p \eta_{t|t_{k-1}}(x, p|x_{t_{k-1}}, p_{t_{k-1}})dx_{t_{k-1}}dp_{t_{k-1}}.$$

Hence, it follows that

$$\int_{t_{k-1}}^{t_k} \int s_\theta(t, x, p)^{\mathrm{T}}M\nabla_p \eta_t(x, p)dxdpdt$$
$$=\int_{t_{k-1}}^{t_k} \int \int \eta_{t_{k-1}}(x_{t_{k-1}}, p_{t_{k-1}})\eta_{t|t_{k-1}}(x, p|x_{t_{k-1}}, p_{t_{k-1}}) \times$$
$$\times s_\theta(t, x, p)^{\mathrm{T}}M\nabla_p \log \eta_{t|t_{k-1}}(x, p|x_{t_{k-1}}, p_{t_{k-1}})dxdpdx_{t_{k-1}}dp_{t_{k-1}}dt$$

so $\zeta^{-1}D_{\mathrm{KL}}(\mathcal{Q}||\mathcal{P}_\theta)$ is equal up to an additive constant $c$ independent of $\theta$ to

$$\sum_{k=1}^{K}\int_{t_{k-1}}^{t_k}\int\int s_\theta(t,x,p)^{\mathrm{T}}Ms_\theta(t,x,p)\eta_{t_{k-1}}(x_{t_{k-1}},p_{t_{k-1}})\eta_{t|t_{k-1}}(x,p|x_{t_{k-1}},p_{t_{k-1}})\mathrm{d}x\mathrm{d}p\mathrm{d}x_{t_{k-1}}\mathrm{d}p_{t_{k-1}}\mathrm{d}t$$

$$-2\sum_{k=1}^{K}\int_{t_{k-1}}^{t_k}\int\int \eta_{t_{k-1}}(x_{t_{k-1}},p_{t_{k-1}})\eta_{t|t_{k-1}}(x,p|x_{t_{k-1}},p_{t_{k-1}})s_\theta(t,x,p)^{\mathrm{T}}M\times$$

$$\times\nabla_p\log\eta_{t|t_{k-1}}(x,p|x_{t_{k-1}},p_{t_{k-1}})\mathrm{d}x\mathrm{d}p\mathrm{d}x_{t_{k-1}}\mathrm{d}p_{t_{k-1}}\mathrm{d}t$$

$$=\sum_{k=1}^{K}\int_{t_{k-1}}^{t_k}\int\int ||s_\theta(t,x,p)-\nabla_p\log\eta_{t|t_{k-1}}(x,p|x_{t_{k-1}},p_{t_{k-1}})||_M^2\eta_{t_{k-1}}(x_{t_{k-1}},p_{t_{k-1}})\times$$

$$\times\eta_{t|t_{k-1}}(x,p|x_{t_{k-1}},p_{t_{k-1}})\mathrm{d}x\mathrm{d}p\mathrm{d}x_{t_{k-1}}\mathrm{d}p_{t_{k-1}}\mathrm{d}t+c$$

Hence, finally we obtain as required

$$D_{\mathrm{KL}}(\mathcal{Q}||\mathcal{P}_\theta)=\zeta\sum_{k=1}^{K}\int_{t_{k-1}}^{t_k}\mathbb{E}\left[||s_\theta(t,x_t,p_t)-\nabla_{p_t}\log\eta_{t|t_{k-1}}(x_t,p_t|x_{t_{k-1}},p_{t_{k-1}})||_M^2\right]\mathrm{d}t+C_2.$$

$$(54)$$

## C   Details of the reverse integrator for the Hamiltonian variant

We take equation (17) and rewrite in the following form.

$$\mathrm{d}\bar{x}_t=-M^{-1}\bar{p}_t\mathrm{d}t,\tag{55}$$
$$\mathrm{d}\bar{p}_t=-\nabla\log\pi_{T-t}(\bar{x}_t)\mathrm{d}t+2\zeta[Ms_\theta(T-t,\bar{x}_t,\bar{p}_t)+\bar{p}_t]\mathrm{d}t+[-\zeta\bar{p}_t\mathrm{d}t+\sqrt{2\zeta}M^{1/2}\mathrm{d}\bar{B}_t].$$

As in [15], we have rewritten the SDE for $\bar{p}_t$ in three distinct components, the last one corresponding to an Ornstein–Ulhenbeck process. The integrator we propose however differs from [15] and will rely on a leapfrog component. In details, we integrate successively over the time interval $\bar{x}\in[t'-\delta,t']\implies x\in[t+\delta,t]$, where $t=T-t'$. This allows us to directly compare random variables with the forward integrator in Section 3.3.

First, we integrate
$$\mathrm{d}\bar{x}_t=-M^{-1}\bar{p}_t\mathrm{d}t,\qquad \mathrm{d}\bar{p}_t=-\nabla\log\pi_{T-t}(\bar{x}_t)\mathrm{d}t\tag{56}$$
by using $(\tilde{x}_{t+\delta},\tilde{p}_{t+\delta})=\Phi_t^{-1}(x_{t+\delta},p_{t+\delta})$ where $\Phi_t^{-1}(x,p)=\Phi_{\mathrm{flip}}\circ\Phi_t\circ\Phi_{\mathrm{flip}}(x,p)$ with $\Phi_{\mathrm{flip}}(x,p)=(x,-p)$. We then integrate

$$\mathrm{d}\bar{x}_t=0,\qquad \mathrm{d}\bar{p}_t=2\zeta[Ms_\theta(T-t,\bar{x}_t,\bar{p}_t)+\bar{p}_t]\mathrm{d}t\tag{57}$$

using $(\hat{x}_t,\hat{p}_t)=(\tilde{x}_{t+\delta},f_\theta(t+\delta,\tilde{x}_{t+\delta},\tilde{p}_{t+\delta}))$ for $f_\theta(t+\delta,\tilde{x}_{t+\delta},\tilde{p}_{t+\delta})=\tilde{p}_{t+\delta}+2\delta\zeta[Ms_\theta(t+\delta,\tilde{x}_{t+\delta},\tilde{p}_{t+\delta})+\tilde{p}_{t+\delta}]$ and finally

$$\mathrm{d}\bar{x}_t=0,\qquad \mathrm{d}\bar{p}_t=-\zeta\bar{p}_t\mathrm{d}t+\sqrt{2\zeta}M^{1/2}\mathrm{d}\bar{B}_t\tag{58}$$

using $(x_t,p_t)=(\hat{x}_t,h\hat{p}_t+\sqrt{1-h^2}\epsilon)$ with $\epsilon\sim\mathcal{N}(0,M)$ for $h=\exp(-\zeta\delta)$.

Note that the $x$ variable only changes in ones of these steps so that $x_t=\hat{x}_t=\tilde{x}_{t+\delta}$, allowing us to eliminate the redundant terms. We can also eliminate $\hat{p}_t$ by substitution. We therefore need only to work in terms of the triple $(x_t,p_t,\tilde{p}_{t+\delta})$ at each combined time step. The conditional distribution for these remaining random variables is then:

$$B_k^\theta(x_t,p_t,\tilde{p}_{t+\delta}|x_{t+\delta},p_{t+\delta})=\delta_{\Phi^{-1}(x_{t+\delta},p_{t+\delta})}(x_t,\tilde{p}_{t+\delta})\mathcal{N}(p_t;hf_\theta(t+\delta,x_t,\tilde{p}_{t+\delta}),(1-h^2)M)$$

$$(59)$$

where $\delta_{\Phi^{-1}(x_{t+\delta},p_{t+\delta})}(x_t,\tilde{p}_{t+\delta})$ is a Dirac-delta centred on $\Phi^{-1}(x_{t+\delta},p_{t+\delta})$. From the main text we note that the forward Hamiltonian integrator has the conditional distribution:

$$F_{k+1}(\tilde{p}_{t+\delta},x_{t+\delta},p_{t+\delta}|x_t,p_t)=\delta_{\Phi(x_t,\tilde{p}_{t+\delta})}(x_{t+\delta},p_{t+\delta})\mathcal{N}(\tilde{p}_{t+\delta};hp_t,(1-h^2)M).\tag{60}$$

We again transition to discretized notation with $\delta := T/K$, and $k = 0, ..., K$. Bringing together all the remaining random variables at each time step we have:

$$Q(\mathbf{x}, \mathbf{p}) = \pi_0(x_0)\mathcal{N}(p_0; 0, M) \prod_{k=0}^{K-1} F_{k+1}(x_{k+1}, p_{k+1}, \tilde{p}_{k+1}|x_k, p_k),$$

$$\Gamma_\theta(\mathbf{x}, \mathbf{p}) = \gamma(x_K)\mathcal{N}(p_K; 0, M) \prod_{k=0}^{K-1} B_k^\theta(x_k, p_k, \tilde{p}_{k+1}|x_{k+1}, p_{k+1}).$$

We are interested in the unnormalized importance weight $w_\theta(\mathbf{x}, \mathbf{p}) = \Gamma_\theta(\mathbf{x}, \mathbf{p})/Q(\mathbf{x}, \mathbf{p})$. We note that the division of delta functions is in general ill-defined and in our case should be interpreted formally in terms of the action of the volume preserving Leapfrog integrator flows $\phi_t$, as we describe in the main text. Further although the numerator and denominator in the definition of $w_\theta(\mathbf{x}, \mathbf{p})$ do not have density with respect to Lebesgue measure, $\Gamma_\theta(\mathbf{x}, \mathbf{p})$ has density with respect to $Q(\mathbf{x}, \mathbf{p})$ and therefore $w_\theta(\mathbf{x}, \mathbf{p})$ may be interpreted as the resulting Radon-Nikodym derivative. The explicit expression for $\log w_\theta(\mathbf{x}, \mathbf{p})$ in equation (21) of the main text then follows.

We now show informally how maximizing the corresponding ELBO $\mathbb{E}_Q[\log w_\theta(\mathbf{x}, \mathbf{p})]$ corresponds approximately to minimizing the score matching loss given in Proposition 3 for $\delta \ll 1$. We restrict the derivation to $M = I$ for simplicity. For pedagogical reasons, it is beneficial here to get back to the continuous time notations and recall that we use $k$ corresponds to $t_k = k\delta$. From direct calculations, maximizing the ELBO is equivalent to minimizing

$$J(\theta) = \sum_{k=0}^{K-1} \mathbb{E}_Q \left[||p_{k\delta} - hf_\theta(k\delta, x_{k\delta}, \widetilde{p}_{(k+1)\delta})||^2\right]$$

where

$$\mathbb{E}_Q \left[||p_{k\delta} - hf_\theta(k\delta, x_{k\delta}, \widetilde{p}_{(k+1)\delta})||^2\right] = \mathbb{E}_Q \left[||p_{k\delta} - h\left(\widetilde{p}_{(k+1)\delta} + 2\delta\zeta[s_\theta(k\delta, x_{k\delta}, \widetilde{p}_{(k+1)\delta}) + \widetilde{p}_{(k+1)\delta}]\right)||^2\right]$$

Note that $h = \exp(-\delta\zeta) \approx 1 - \delta\zeta$ so we obtain

$$\mathbb{E}_Q \left[||p_{k\delta} - hf_\theta(k\delta, x_{k\delta}, \widetilde{p}_{(k+1)\delta})||^2\right]$$
$$\approx \mathbb{E}_Q \left[||p_{k\delta} - (1 - \delta\zeta)\left(\widetilde{p}_{(k+1)\delta} + 2\delta\zeta[s_\theta(k\delta, x_{k\delta}, \widetilde{p}_{(k+1)\delta}) + \widetilde{p}_{(k+1)\delta}]\right)||^2\right]$$
$$\approx \mathbb{E}_Q \left[||p_{k\delta} - (1 + \delta\zeta)\widetilde{p}_{(k+1)\delta} - 2\delta\zeta s_\theta(k\delta, x_{k\delta}, \widetilde{p}_{(k+1)\delta})||^2\right]$$

by neglecting terms of order $\delta^2$. Now we try to further understand the asymptotic when $\delta \to 0$. We have that

$$(x_{k\delta}, \widetilde{p}_{(k+1)\delta}) = \Phi_{k\delta}^{-1}(x_{(k+1)\delta}, p_{(k+1)\delta}).$$

Now as we use for $\Phi_{k\delta}$ a leapfrog-type integrator, we do have

$$\Phi_{k\delta}^{-1}(x', p') = \Phi_{\text{flip}} \circ \Phi_{k\delta} \circ \Phi_{\text{flip}}(x', p')$$

where $\Phi_{\text{flip}}(x, p) = (x, -p)$ and, for $\delta \to 0$, we have

$$\Phi_{k\delta}(x, p) \approx (x + \delta p, p - \delta\nabla E_{k\delta}(x)),$$

where $\pi_{k\delta}(x) \propto \exp(-E_{k\delta}(x))$. So we have for

$$\Phi_{k\delta}^{-1}(x', p') = \Phi_{\text{flip}} \circ \Phi_t \circ \Phi_{\text{flip}}(x', p')$$
$$= \Phi_{\text{flip}} \circ \Phi_{k\delta}(x', -p')$$
$$= \Phi_{\text{flip}}(x' - \delta p', -p' - \delta\nabla E_{k\delta}(x'))$$
$$= (x' - \delta p', p' + \delta\nabla E_{k\delta}(x')).$$

It follows that

$$\mathbb{E}_Q \left[||p_{k\delta} - (1 + \delta\zeta)\widetilde{p}_{(k+1)\delta} - 2\delta\zeta s_\theta(k\delta, x_{k\delta}, \widetilde{p}_{(k+1)\delta})||^2\right]$$
$$\approx \mathbb{E}_Q \left[||p_{k\delta} - (1 + \delta\zeta)(p_{(k+1)\delta} + \delta\nabla E_{k\delta}(x_{(k+1)\delta})) - 2\delta\zeta s_\theta(k\delta, x_{(k+1)\delta}, p_{(k+1)\delta})||^2\right]$$
$$\approx \mathbb{E}_Q \left[||p_{k\delta} - p_{(k+1)\delta} - \delta(\zeta p_{(k+1)\delta} + \nabla E_{k\delta}(x_{(k+1)\delta}) + 2\zeta s_\theta(k\delta, x_{(k+1)\delta}, p_{(k+1)\delta}))||^2\right].$$

Now we have

$$\frac{1}{4\zeta\delta}\mathbb{E}_Q\left[||p_{k\delta} - p_{(k+1)\delta} - \delta(\zeta p_{(k+1)\delta} + \nabla E_{k\delta}(x_{(k+1)\delta}) + 2\zeta s_\theta(k\delta, x_{(k+1)\delta}, p_{(k+1)\delta}))||^2\right]$$

$$=\delta\zeta\mathbb{E}_Q\left[||s_\theta(k\delta, x_{(k+1)\delta}, p_{(k+1)\delta}) - \frac{1}{2\zeta\delta}(p_{k\delta} - p_{(k+1)\delta} - \delta(\zeta p_{(k+1)\delta} + \nabla E_{k\delta}(x_{(k+1)\delta})))||^2\right]$$

$$\approx\delta\zeta\mathbb{E}_Q\left[||s_\theta(k\delta, x_{(k+1)\delta}, p_{(k+1)\delta}) - \frac{1}{2\zeta\delta}(p_{k\delta} - p_{(k+1)\delta} - \delta(\zeta p_{k\delta} + \nabla E_{k\delta}(x_{k\delta})))||^2\right]$$

$$=\delta\zeta\mathbb{E}_Q\left[||s_\theta(k\delta, x_{(k+1)\delta}, p_{(k+1)\delta}) - \nabla_{p_{(k+1)\delta}}\log F(p_{(k+1)\delta}|p_{k\delta}, x_{k\delta})||^2\right], \tag{61}$$

where $F(p_{(k+1)\delta}|p_{k\delta}, x_{k\delta}) = \mathcal{N}(p_{(k+1)\delta}; (1 - \delta\zeta)p_{k\delta} - \delta\nabla E_{k\delta}(x_{k\delta}); 2\zeta\delta I)$ and the joint distribution $F(x_{(k+1)\delta}, p_{(k+1)\delta})|p_{k\delta}, x_{k\delta}) = \delta_{x_{k\delta} - \delta p_{k\delta}}(x_{(k+1)\delta})F(p_{(k+1)\delta}|p_{k\delta}, x_{k\delta})$ is an Euler approximation of the forward transition of the underdamped Langevin dynamics. Now we expect similarly $\eta_{(k+1)\delta|k\delta}(x_{(k+1)\delta}, p_{(k+1)\delta}|x_{k\delta}, p_{k\delta}) \approx \eta_{(k+1)\delta|k\delta}(p_{(k+1)\delta}|x_{k\delta}, p_{k\delta})\eta_{(k+1)\delta|k\delta}(x_{(k+1)\delta}|x_{k\delta}, p_{k\delta})$ for $\delta \ll 1$ and (61) is thus an approximation of the score matching loss.

## D   Diffusion processes: SGM, Langevin and AIS

We provide here a more detailed discussion between the similarities and differences between the diffusion process considered for SGM and the proposed approach. We first recall some basic elements of diffusion processes. Consider the diffusion $(x_t)_{t\in[0,T]}$ on $\mathbb{R}^d$

$$\mathrm{d}x_t = f(t, x_t)\mathrm{d}t + \sqrt{2}\mathrm{d}B_t, \qquad x_0 \sim q_0, \tag{62}$$

where $(B_t)_{t\in[0,T]}$ is standard multivariate Brownian motion and $q_0$ is the initial distribution. The law $q_t$ of $x_t$ induced by this diffusion satisfies the Fokker–Planck–Kolmogorov equation

$$\frac{\partial q_t(x)}{\partial t} = -\nabla \cdot [f(t, x)q_t(x)] + \Delta q_t(x) \tag{63}$$

where $\nabla \cdot [f(t, x)q_t(x)] = \sum_{i=1}^d \frac{\partial[f_i(t,x)q_t(x)]}{\partial x_i}$ and $\Delta q_t(x) = \sum_{i=1}^d \frac{\partial^2 q_t(x)}{\partial x_i^2}$ denote the divergence and Laplacian operators; see e.g. [32].

For SGM, we define the following Ornstein–Ulhenbeck process on $\mathbb{R}^d$ which corresponds to using

$$f_{\mathrm{OU}}(t, x) = -x \tag{64}$$

where $(B_t)_{t\in[0,T]}$ is standard multivariate Brownian motion and $q_0$ is the data distribution. This is also known in the SGM literature as the variance preserving diffusion [48]. This process adds noise progressively to the complex data distribution and converges geometrically fast to its invariant distribution which is the standard multivariate Gaussian $\pi_{\mathrm{OU}}(x) = \mathcal{N}(x; 0, I)$ (see e.g. [12]) verifying indeed that the r.h.s. of (63) satisfies

$$-\nabla \cdot [f_{\mathrm{OU}}(t, x)\pi_{\mathrm{OU}}(x)] + \Delta\pi_{\mathrm{OU}}(x) = 0.$$

More generally, a time-homogeneous Langevin diffusion to sample from a target distribution $\pi$ is of the form

$$f_{\mathrm{Lgv}}(t, x) = \nabla \log \pi(x). \tag{65}$$

It is easily check that $\pi$ is indeed an invariant distribution as the r.h.s. of (63) satisfies

$$-\nabla \cdot [f_{\mathrm{Lgv}}(t, x)\pi(x)] + \Delta\pi(x) = 0,$$

so that $q_t = \pi$ for all $t$ if $q_0 = \pi$. Moreover, $q_t$ converges to $\pi$ whatever being $\pi_0$; see e.g. [42]. However, obtaining sharp quantitative bounds for complex $\pi$ is a more difficult task than for $\pi_{\mathrm{OU}}(x) = \mathcal{N}(x; 0, I)$ in general.

In the context of this paper, the "forward" diffusion process we define is a *time-inhomogeneous* Langevin algorithm

$$f_{\mathrm{AIS}}(t, x) = \nabla \log \pi_t(x), \tag{66}$$

with $q_0 = \pi_0$ an easy-to-sample distribution and $(\pi_t)_{t\in[0,T]}$ a non-constant curve of distributions such that $\pi_T = \pi$. So, contrary to SGM, which starts from a complex distribution and moves

towards a simple distribution, we start here from a simple distribution and moves towards a complex distribution.

In this scenario, even in $q_0 = \pi_0$ then we do not have $q_t = \pi_t$ as the diffusion always lags behind its stationary distribution at time $t$. However, quantitative results measuring the discrepancy between the law of $x_T$ and $\pi$ for such annealed diffusions have been obtained; see e.g. [20, 17, 50]. For this discrepancy to be small, one requires $\pi_t$ to vary slowly over time.

# E    Experimental Details

In all experiments, we sweep over diffusion time, number of steps, step-sizes and whether to learn them, and the annealing schedule. We identify the best parameters for each sampler individually on a validation set and then re-run these methods using 5 different seeds to obtain error bars on test set performance. All experiments were executed on 8 GPUs for parallelized training and a single instance of our most expensive experiment (VAE) takes under 3 hours including evaluation. Experiments are implemented in JAX [6] using the DeepMind JAX ecosystem [4].

## E.1    Sampler parameterization

For all models, the step size was learned via a function $\epsilon_\theta(t)$ which is a 2-layer neural network with 32 hidden units, followed by a scaled sigmoid function which constrains $\epsilon_\theta(t) < 0.25$. As in prior work [18] we found this alleviated some instabilities in training.

When learning the annealing schedule, we parameterize an increasing sequence of $T$ steps using unconstrained parameters $b_t$ (initialized to the same constant). We map these to our annealing schedule with

$$\beta_t = \frac{\sum_{t' \leq t} \sigma(b_{t'})}{\sum_{t'=1}^{T} \sigma(b_{t'})} \tag{67}$$

where we fix $\beta_0 = 0$ and $\sigma$ is the sigmoid function. This ensures that $\beta_0 = 0$, $\beta_K = 1$, and $\beta_t < \beta_{t'}$ when $t < t'$.

For UHA [18], we also learn the momentum refreshment parameter $\eta \in (0, 1)$. We parameterize this with a parameter $u$ and define $\eta = .98\sigma(u) + .01$ to keep the values in the range $(.01, .99)$ which we found alleviated sone training instabilities.

## E.2    Score model parameterization

We parameterize our score model $s_\theta(t, x)$ using an MLP residual network. We first project the $x$ to dim $d_h$ using a linear layer and embed discrete time steps $t$ to dim $d_t$ using a learned embedding map. We then apply $k$ residual blocks.

Each block begins with a layer norm [3] operation followed by a nonlinearity. We project the hidden representation to dim $2 \cdot d_h$ using a linear layer, project the embedding of $t$ to dim $2 \cdot d_h$ using another linear map and add them together. We then apply another nonlinearity and then project the back to $d_h$ using another linear layer. We use the swish nonlinearity [40] throughout.

To ensure our ELBO is initialized to a reasonable value we *warm start* it so that at initialization, the score model outputs the standard AIS backward kernels. For the ULA version of our approach we do this by defining a score model $\tilde{s}_\theta(t, x)$ as explained above (but set the final layer weights to 0 at initialization) and define:

$$s_\theta(t, x) = \tilde{s}_\theta(t, x) + \nabla_x \log \gamma_t(x). \tag{68}$$

For the UHA version we parameterize a score model $\tilde{s}_\theta(t, x, p)$ and define:

$$s_\theta(t, x, p) = \tilde{s}_\theta(t, x, p) - M^{-1}p. \tag{69}$$

In both cases we found this led to much faster convergence and better results overall.

### E.3 Hyper-parameters

In all experiments we use $k = 3$ residual blocks in our score network. For our Gaussian experiments we set $d_h = 512$ and $d_t = 16$. All models are trained with the Adam optimizer [30] with learning rates $0.001$ and $0.0001$ and up to 300k iterations with a batch size of 128. For static targets, we produce an estimate of $\log Z$ using 16,384 importance samples. As these methods produce a stochastic lower-bound on $\log Z$ we report the result from the hyper-parameter setting which gives the largest $\log Z$ estimate.

The VAE experiment uses architectures described in [8], which consists of encoder and decoder MLPs with two hidden layers with 200 units each, $\tanh$ activations, and 50 latent dimensions. In contrast to [18], we found this architecture work better than the one described in [18], especially when trained for more iterations. We chose the best performing models and their hyperparameters by monitoring validation performance during training. We report performance of the best combination on the full test set, for each model respectively. The best performance were reached with a matched the number of sampler steps between ULA/ULA-MCD and UHA/UHA-MCD – 64 and 32 respectively.

## F  Additional Results

### F.1  Static Targets

Here we present additional results on a $\mathcal{N}(10, I)$ target, a $\mathcal{N}(0, 0.1I)$ target, and a Laplace$(0, I)$ target. Results can be found in Tables 4, 5, and 6.

| Sampler | ULA | | UHA | | ULA-MCD | | UHA-MCD | |
|---|---|---|---|---|---|---|---|---|
| # steps | 64 | 256 | 64 | 256 | 64 | 256 | 64 | 256 |
| Dim-20 | -46.75 $\pm$ 0.69 | -6.23 $\pm$ 0.91 | 0.0002 $\pm$ 0.0008 | 0.0002 $\pm$ 0.0004 | -0.017 $\pm$ 0.020 | 0.0034 $\pm$ 0.0055 | -0.0005 $\pm$ 0.0007 | 0.0000 $\pm$ 0.0005 |
| Dim-200 | -752.25 $\pm$ 2.43 | -160.60 $\pm$ 2.03 | 0.0003 $\pm$ 0.0026 | -0.0005 $\pm$ 0.0007 | -4.74 $\pm$ 1.20 | -0.019 $\pm$ 0.047 | 0.0008 $\pm$ 0.0032 | 0.0060 $\pm$ 0.0084 |
| Dim-500 | -1999.40 $\pm$ 18.49 | -455.20 $\pm$ 7.90 | 0.0006 $\pm$ 0.0007 | -0.0008 $\pm$ 0.0012 | -21.62 $\pm$ 1.64 | -0.29 $\pm$ 0.16 | 0.0030 $\pm$ 0.0063 | 0.0099 $\pm$ 0.0050 |

Table 4: $\log Z$ estimates for a $\mathcal{N}(10, I)$ target. Averages and standard errors over 3 seeds.

| Sampler | ULA | | UHA | | ULA-MCD | | UHA-MCD | |
|---|---|---|---|---|---|---|---|---|
| # steps | 64 | 256 | 64 | 256 | 64 | 256 | 64 | 256 |
| Dim-20 | -1.58 $\pm$ 0.42 | -0.27 $\pm$ 0.13 | -4.46 $\pm$ 0.93 | -0.41 $\pm$ 0.51 | 0.0095 $\pm$ 0.0155 | 0.0057 $\pm$ 0.0052 | 0.0038 $\pm$ 0.0273 | 0.0002 $\pm$ 0.0120 |
| Dim-200 | -166.23 $\pm$ 5.15 | -58.33 $\pm$ 1.99 | -228.79 $\pm$ 3.64 | -74.62 $\pm$ 1.60 | -4.53 $\pm$ 0.96 | -0.67 $\pm$ 0.33 | -4.10 $\pm$ 0.98 | -1.47 $\pm$ 0.57 |
| Dim-500 | -545.77 $\pm$ 5.55 | -207.15 $\pm$ 5.73 | -704.96 $\pm$ 7.30 | -247.11 $\pm$ 4.83 | -27.36 $\pm$ 2.23 | -10.11 $\pm$ 1.17 | -29.20 $\pm$ 3.46 | -5.14 $\pm$ 1.50 |

Table 5: $\log Z$ estimates for a $\mathcal{N}(0, 0.1I)$ target. Averages and standard errors over 3 seeds.

| Sampler | ULA | | UHA | | ULA-MCD | | UHA-MCD | |
|---|---|---|---|---|---|---|---|---|
| # steps | 64 | 256 | 64 | 256 | 64 | 256 | 64 | 256 |
| Dim-20 | 0.31 $\pm$ 0.45 | 0.40 $\pm$ 0.63 | -0.0086 $\pm$ 0.1314 | -0.0077 $\pm$ 0.1340 | 0.092 $\pm$ 0.235 | -0.23 $\pm$ 0.02 | 0.0003 $\pm$ 0.1573 | -0.020 $\pm$ 0.141 |
| Dim-200 | -5.40 $\pm$ 0.53 | -4.54 $\pm$ 0.96 | -5.27 $\pm$ 0.18 | -5.28 $\pm$ 0.19 | -5.08 $\pm$ 0.52 | -4.31 $\pm$ 0.66 | -5.30 $\pm$ 0.15 | -5.39 $\pm$ 0.09 |
| Dim-500 | -17.67 $\pm$ 1.91 | -17.25 $\pm$ 0.85 | -17.91 $\pm$ 0.78 | -17.91 $\pm$ 0.78 | -17.64 $\pm$ 1.98 | -15.52 $\pm$ 1.26 | -18.11 $\pm$ 0.70 | -18.11 $\pm$ 0.72 |

Table 6: $\log Z$ estimates for a Student-T target. Averages and standard errors over 3 seeds.

### F.2 Normalizing Flow

In this experiment we train NICE [14] flows which are fitted on downsampled variants of the MNIST dataset at resolutions $7 \times 7$, $14 \times 14$, and $28 \times 28$. All models are trained for 100K steps with a batch size of 128 and then $\log Z$ is estimated using 4096 importance samples.

Results can be seen in Table 7. In the largest setting we can see that UHA outperforms ULA, but our method outperforms both. We note that a Sequential Monte Carlo sampler or AIS with Metropolis-Hastings corrections would be capable of estimating $\log Z$ very close to the true value of 0. We have found that unadjusted samplers required for building a differentiable evidence lower-bound have difficulties sampling from this target distribution. As discussed in the Limitations section of the main paper, we hypothesize that this is due to the fact that we are limited to using a relatively small number of transitions, since backpropagating through these repeated updates can be unstable. We believe this is a limitation of this approach to inference and will impact any approach which utilizes an unadjusted forward sampler. Still, we note that our approach to learning an optimized reversal (MCD) leads to improvements over the standard AIS reversals. We believe addressing these issues to be a key area for future research to focus.

| Dimension | ULA | UHA | MCD (ours) |
|:---:|:---:|:---:|:---:|
| $7 \times 7$ | 0.14 | 0.17 | **0.11** |
| $14 \times 14$ | 13.24 | 15.04 | **6.25** |
| $28 \times 28$ | 141.29 | 82.16 | **23.10** |

Table 7: $\log Z$ estimate absolute error for Normalizing flows.

### F.3 Score Network Ablation

We include additional results exploring the impact of various score-network architectures on performance. We re-run our Gaussian Mixture experiments in dimension 200 using 1) an MLP with residual connections and 2) a standard MLP, both with 1, 2, and 3 layers. We run these experiments with 64 and 128 sampling steps. Results can be seen in Figure 4. We see that more expressive architectures lead to better performance in general. Further, we find that with more sampling steps, the impact of a more expressive score network diminishes. This aligns with intuition as, when using more steps we will use a smaller step size, and the standard AIS reversal becomes a better approximation to the true reversal.

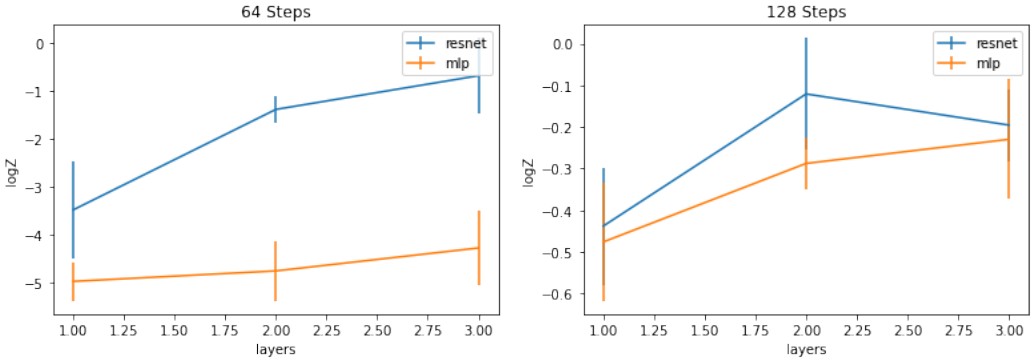

Figure 4: Score network architecture ablation on Gaussian mixture target (main results can be found in Table 1). Left: results with 64 sampling steps, right: results with 128 sampling steps.