# OpenReview forum: "Score-Based Diffusion meets Annealed Importance Sampling"
_NeurIPS.cc/2022/Conference — NeurIPS 2022 Accept_

### Official Review · Reviewer_aT45 · 2022-06-30

**Rating:** 7
**Confidence:** 4
**Soundness:** 4 excellent
**Presentation:** 3 good
**Contribution:** 3 good

**Summary:**

The key problem this work addresses is the suboptimality of the extended target distribution that is widely used in current annealed importance sampling (AIS) methods. The authors, building on previous literature, show that the custom choice of the backward Markov kernel of the unnormalized extended target is comfortable but suboptimal in terms of variance.
To overcome such limitations, the authors extend some key observations that have been made in the literature of diffusion processes used for generative modeling (e.g. Sohl-Dickstein et al., 2015), and draw a connection between forward/backward diffusion and forward/backward transition kernels used in importance sampling. Armed with this realization, the authors show that AIS can be studied through the lenses of forward and backward diffusion SDEs: the forward diffusion begins in a simple to sample from distribution, and gradually transforms it into the target distribution (or an extension thereof), while the backward diffusion corresponds to the (hypothetical) process of transforming the target to a simple distribution.
In addition, and as an extension to a previous paper that appeared in a recent workshop, the authors consider Hamiltoinan dynamics, and derive a similar connection to AIS, but with improved performance due to a higher fidelity representation of the paths taken by the Hamiltonian SDEs, when compared to first order dynamics.
For practical reasons, the authors produce discretized versions of the dynamics they study, and implement numerical integration schemes for the simulation of the SDEs, where necessary.
Numerical results on both static distributions and on an instance of a variational autoencoder indicate dramatic performance improvements of the proposed approach when compared to vanilla and Hamiltonian vanilla versions of AIS.

**Questions:**

Here are some questions, but these are somehow superficial as I didn't find any major flaws in the paper.

1- I think there is a typo in eq 12: the term $ 2 \nabla \log s_{\theta} ( \cdot ) $ should read $ 2 s_{\theta} ( \cdot ) $

2- I may have made mistakes in my own derivations, but isn't there a typo in eq 6 for the backward transition kernel? In the numerator you should have $q_{k+1}(x_k)$ and not $q_{k}(x_k)$? I'm not 100% sure here

3- May I suggest to improve the notation? I find it difficult to follow when $x, x'$ are used in place of $x_{k+1}, x_k$

4- Would it make sense to have an additional section in the appendix to clarify not only the similarities but also the differences w.r.t. score-based diffusion models? See my comment on the weaknesses above.

== Post Rebuttal ==
Thanks for your answers to the questions and comments above. I have modified my score, raising it to an accept/7.

**Limitations:**

I think they did address limitations appropriately.

**Strengths And Weaknesses:**

Strengths:
* This is a very important topic, and the performance gains from the proposed method are substantial
* I liked the idea of connecting the dots between AIS and (score-based) diffusion models
* The extension to Hamiltoinan dynamics is interesting, and a nice complement to [1]

Weaknesses:
* I think the exposition of this paper should be improved: the connection with diffusion models is somehow reversed, as in that literature the forward process is going from the target distribution to a simple one, and the backward process is doing the opposite, whereas in this paper it is the other way around. This could be discussed in the appendix. Also, I would study the "steady state" distribution of the forward process defined in eq 10, by means of Fokker-Plank equations, to show that indeed the forward process moves an "easy" distribution to the target one. This can be helpful for a reader familiar with diffusion models, whereby you can demonstrate that for variance preserving methods, the process converges to isotropic Gaussian.
To put things differently: a reader with familiarity in diffusion models knows that the forward process "destroys" data, whereas in this work it is the other way around.
* experiments only compare the proposed method to baseline AIS and to Hamiltonian AIS (which I appreciated a lot). However, this work could be cast under the umbrella of energy based models and density estimation. Then, would it make sense to compare it to such line of work?
* there exist a workshop paper presenting preliminary ideas that are extended in this work. The main difference here is the treatment of Hamiltonian dynamics, and the experiments with the variational autoencoder. If accepted, this paper could maybe give more space to the Hamiltonian part, and most importantly, to the excellent results obtained. This is the main reason why, in its current form, the key contribution of this paper is somehow diluted, and seem a repetition: I like this work, and I would be happy to raise the score I gave which is only tainted by this discussed weakness

[1] Tim Dockhorn and Arash Vahdat and Karsten Kreis, Score-Based Generative Modeling with Critically-Damped Langevin Diffusion, ICLR 2022

---

> ### Author Response · Authors · 2022-08-02
> **Thank you, here is our reply**
>
> We would like to thank the reviewer for their high quality and comprehensive review. We are pleased to hear that the reviewer recognizes the importance of the topic.
>
>
> Workshop paper: It is our understanding that the publication of a related workshop paper should have no bearing on the score of the current submission as pointed out in the following entry of the [reviewer FAQ](https://docs.google.com/document/d/1uKc89EcSh0hZJo8KJtY0Lh6ZTWPLZvqWHNhR8WsNDoQ/edit):
> “Q: What if I’ve seen similar work in a NeurIPS/ICML workshop?
> A: We allow work that has been submitted to non-archival workshops to be submitted to NeurIPS. To maintain anonymity, do not mention the workshop paper in your review.”
>
> Relationship to diffusion models: We thank the reviewer for this excellent suggestion. We agree that it would help the exposition of the paper to further elaborate the contrast of diffusing  target->noise vs noise->target. We have revised the paper in this direction, had some comments in the main text and added a discussion in Appendix D.
>
> Comparisons to EBMs / density estimation:
> We were not completely sure what this comment referred to. We believe the reviewer was asking why we did not apply our method towards training energy-based models. If this was indeed the case, we provide the following response:
>
> The main focus of this work is a method for partition function estimation. One very appealing application for such methods would be in training energy-based models. There exists prior work using inference methods such as ours for this task [1, 3]. In particular some prior works [2, 4] have utilized inference methods based on MCMC-sampling with some success. There arises additional complexity when working EBMs due to difficulty in evaluation. EBM evaluation remains an open research problem and it is very difficult to obtain tight enough bounds on likelihood to reliably compare the performance of models trained in different ways. For this reason we decided to focus our evaluation on inference problems with easy-to-compute evaluation metrics. We are very excited to apply our method towards EBM training in the future, but we felt this to be slightly outside the context of the current work and applying our method to this task should be the focus of a subsequent publication.
>
> We hope this answers your question. If we misunderstood, we apologize, and if possible could you please elaborate more to improve our understanding?
>
> * [1] https://arxiv.org/abs/2010.04230
> * [2] https://proceedings.neurips.cc/paper/2019/hash/767d01b4bac1a1e8824c9b9f7cc79a04-Abstract.html
> * [3] https://arxiv.org/pdf/1901.08508.pdf
> * [4] https://arxiv.org/pdf/2006.06897.pdf
>
>
> Questions:
> 1) Thanks for pointing this out. This is indeed a typo. We have corrected it.
> 2) This is correct as written. This can be checked given $q_{k+1}(x_{k+1})=\int q_k(x_k)F_{k+1}(x_{k+1}|x_k)dx_k$.
> 3) Thanks for pointing this out, we have suppressed the use of $x,x’$ and this indeed improves clarity.
> 4) Absolutely, this is a great suggestion and we have added a few sentences in the main paper and a section in the Appendix (Appendix D) to discuss this.

---

> > ### Comment · Reviewer_aT45 · 2022-08-08
> > **Thank you**
> >
> > Dear authors,
> > thanks for your message, and apologies for being a bit late with my comments.
> > All looks good to me, I will raise my score and cross fingers for this paper to make it to the conference.
> >
> > Cheers

---

### Official Review · Reviewer_xpLj · 2022-07-10

**Rating:** 6
**Confidence:** 3
**Soundness:** 3 good
**Presentation:** 3 good
**Contribution:** 2 fair

**Summary:**

Annealed importance sampling is one of the most effective methods for marginal likelihood estimation. The commonly used extended target distribution is simple but sub-optimal. The paper leverage the recent progress in score-based generative modeling to approximate the optimal extended target distribution for annealed importance sampling and give the discretization of Langevin and Hamiltonian dynamics. On a number of synthetic distributions and variational auto-encoders, the proposed method obtains better performance compared to existing methods.

**Questions:**

* Considering unadjusted algorithms are used, will the error in score estimation brings bias in the sampling? If so, is it able to quantify the bias in terms of the estimation error?

* What is the training time of the score model? How is it compared to the sampling time?

**Limitations:**

The paper mentions some limitations of the proposed methods. I think it misses two important problems:
1. The method requires learning a score model, which is time consuming and hard. This factor reduces the efficiency or even correctness of the proposed method.
2. The estimation error in the score model may lead to bias in sampling, and the paper lacks a quantification of the potential bias.

**Strengths And Weaknesses:**

* The writing is of high quality. The paper gives a concise review of the annealed importance sampling and illustrates the limitations clearly via two examples. Then, the paper provides necessary details in deriving the Langevin and Hamiltonian algorithms and makes them easy to read.

* As a sampling algorithm, the effectiveness of the proposed method needs further consideration. In particular, the effectiveness of the proposed method relies on the qualities of the learned score function. However, learning the score models is not easier than estimating the partition functions. Whether the proposed method can apply well to hard distributions is still doubtful. It will be helpful if the paper can provide an ablation study that investigates the relationship between the quality of the learned score functions and the MCD samplers.

* Although I have some reservations about the effectiveness of the current algorithm due to the reason mentioned above, its contribution to further studies on extended target distributions should be recognized. For example, when the energy function is learned via a score-based model and is associated with a score function in nature, the proposed method could be very helpful.

* The evaluation in the experiment is not fair. We can run ULA and UHA directly, but we need to learn the score functions to run ULA-MCD and UHA-MCD. The paper should also report the learning time and the sampling time such that the readers can have better judgments.

---

> ### Author Response · Authors · 2022-08-02
> **Thank you, here is our reply**
>
> We thank the reviewer for their thoughtful comments. We are pleased to hear that they find the writing of high quality and easy to read.
>
> Ablations for score model: We agree with the reviewer that an ablation of the expressiveness of the score model would be an excellent addition to the paper. We have added an additional table in the appendix (Figure 4) that explores the impact of the score model architecture. We notice improvements with more expressive score models. The impact of the score model architecture is larger when using a smaller number of sampling steps. This aligns with intuition since, as the step size increases, the distributions which the score model needs to approximate becomes less smooth.
>
> Score-learning vs partition functions: The reviewer states that learning the score is not easier than estimating partition functions. In general scenarios it would, but here we are estimating the scores of a diffusion process we can sample from easily and this boils down to solving a regression problem. In the context of score-based generative modeling, one similarly estimates the scores of a diffusion process (albeit a diffusion different from ours) one can sample from easily. The explosion of work in this area shows that the scores of very complex distributions can be learned efficiently.
>
> Bias: You are correct that unadjusted samplers will lead to bias in the resulting samples. In the context of time-homogeneous ULA, there has been much recent work quantifying this bias and we have included some references. However, note here that these samplers are used to build the proposal distribution Q (Eq 2) in an importance sampling scheme that targets the distribution of interest. So in particular, we do obtain an unbiased estimate $w(x_{0:K})$ of the evidence (Eq 4) from which we can build directly an ELBO by taking the expectation of the logarithm of this estimate. Such an approach is also adopted in all previous work on the topic; see e.g. Wu et al., 2020; Thin et al., 2021, Geffner, T. and Domke, J. (2021), and Zhang, G., Hsu, K., Li, J., Finn, C., and Grosse, R. (2021).
>
> Fair evaluation & training time: The reviewer raises a good point that learning the score while sampling comes with an additional computational cost. The natural way to “match” the computational budget of ULA/UHA with that of our MCD extensions is to increase the number of integration steps of the baselines, which translates into the number of target gradient evaluations. The total computational load of our MCD extensions is roughly 2x that of the baselines (see Table 3 in the revised version of paper). With that in mind, we refer the reviewer to Tables 1 & 2, where our method performs favorably when using a quarter (64 instead 256 sampler steps) of the total gradient target evaluations.

---

> > ### Comment · Reviewer_xpLj · 2022-08-03
> > **Thanks for your reply**
> >
> > The authors have answered most of the questions. But I am still confused about two of them.
> > * For the ablation study: I think the author should add the curve of the baselines in Figure 4. The interesting part of this ablation is to quantify how much expressiveness the score model needs so as to outperform the baselines. Intuitively, if the target distribution becomes more complex, we will need a more flexible score model to match the score. I am curious if there is a threshold of the score matching error such that only the score models better than the threshold can improve the efficiency of the baselines. And if there exists such a threshold, how does it change when the target distribution becomes more difficult (e.g. EBMs on harder datasets)?
> > * For evaluation & training time: I am confused about the statement "The total computational load of our MCD extensions is roughly 2x that of the baselines (see Table 3 in the revised version of paper)" Does this ratio only applies to amortized VI or it also applies to the static sampling in section 4.1? Does the computational load of MCD include the training time of score functions and VAE? Does the computational load of MCD include the training time of VAE? Since Section E.3 reports that 16384 samples are used to estimate $\log Z$, I think it will be a good explanation if the paper can report how many samples are used to train the score model.

---

> > > ### Author Response · Authors · 2022-08-04
> > > **Clarifications**
> > >
> > > * We agree that it would be useful to include the baselines in the ablation results in Figure 4. Note that this would be in the form of horizontal lines for each baseline, with values taken from Table 1 (for the left plot in Fig 4). The values are ULA and UHA are -85.62 and -8.20 respectively, compared to the range [-5,0] of MCD. We will add this in the final appendix. This makes it clear that our MCD extension outperforms these baselines drastically even for smaller score networks with a supposedly larger score error. Clearly, as the score network error becomes larger, our method performs worse. E.g. MCD-ULA with the smallest MLP reaches a log Z of ~ -5, compared to ~-8 of UHA without MCD. As argued above though, noise-conditional score estimation is known to work quite well, even on the complex datasets considered in score-based generative models, and the additional computational effort seems to be justified in the examples shown.
> > >
> > > * We only included time results for the VAE target as they are slightly easier to measure accurately as the target evaluation is more expensive and therefore overhead from function calls etc distorts the timings less. But yes, the ratio also applies to the static targets from Sec 4.1. E.g. on the 200-dim Gaussian mixture target from Table 1, ULA and ULA-MCD take ~0.1s and ~0.24s respectively. We will add the timings (with the caveat that they are less accurate for fast to evaluate targets) in the final version of the appendix. Yes, the iteration time in the VAE experiment includes a gradient step on the end-to-end differentiated ELBO (including VAE encoder/decoder, and score model). We highlight that the increased computational time from the MCD method comes from backproping through the score network. Both the baselines and MCD need to backprop through the VAE parameters.
> > >
> > > * The batch size (number of samples used to estimate elbo/score) during training was either 256 or 512. We forgot to add those numbers and will update the section in an updated version. We use substantially more samples at test time to obtain tighter estimates (done for all methods).
> > >
> > > We hope this answers your questions.

---

> > > > ### Comment · Reviewer_xpLj · 2022-08-06
> > > > **Thanks for the clarification**
> > > >
> > > > Thank the authors to make the clarification. They answer my questions.
> > > >
> > > >  I believe the authors will add the results to their final version. Overall, I think this is a good paper.

---

### Official Review · Reviewer_KR2v · 2022-07-11

**Rating:** 7
**Confidence:** 3
**Soundness:** 3 good
**Presentation:** 3 good
**Contribution:** 4 excellent

**Summary:**

This paper proposes an improved Annealed Importance Sampling (AIS) for score-based diffusion models. The author argues that past papers on AIS have predominantly focused on modifying the intermediate distributions or the Markov kernel but have overlooked the suboptimal nature of the extended target distribution. Thus, the paper integrates a new AIS procedure in combination with recent works to approximate the extended target distribution.

**Questions:**

- Were there any experiments done on ULA-MCD and UHA-MCD on more complex distributions? I understand estimating the normalizing constant is very difficult, but a single experiment would be nice to see.
- Why do we need applications for amortized inference? Also, why not experiment on more datasets?
- I am not very familiar with AIS, are differentiable AIS and ULA and UHA the current SOTA benchmarks for likelihood estimation?
- How would you provide a solution for the numerical instability in optimizing due to the unadjusted Langevin and Hamiltonian sampler? This seems like an imperative issue that needs to be addressed.

**Limitations:**

The authors transparently addresses the limitations of their work and have a section dedicated to limitations.

**Strengths And Weaknesses:**

Strengths:
- The paper is well motivated and theoretically proves the correctness of the proposed methods.
- Overall, the paper is well written and explained useful insight on the limitations on the Markov kernel:
  - E.g. the illustration of the suboptimality of a homogeneous MCMC chain give good intuition on the current problem of AIS.

Weaknesses:
- While experimental results show improvement, it would be more convincing if there were more real-world dataset experiments (other than binarized MNIST).

Minor Suggestions:
- Why not bold the best log Z estimate on Table 1 & 2?

---

> ### Author Response · Authors · 2022-08-02
> **Thank you, here is our reply**
>
> We appreciate the review and are pleased that the reviewer finds the paper well motivated and well written.
>
> Amortized Inference: AIS methods are traditionally used to estimate normalizing constants for fixed distributions. There is, however, a growing body of research that studies the benefits of AIS in the context of generative models, i.e. for posterior inference (Thin et al, 2021, Geffner & Domke 2022, Zhang et al., 2022). For deep generative models, this typically means dealing with an amortized inference process and requires end-to-end differentiability. For our work, this requires further conditioning of the score estimator on the current inputs of the model. We demonstrate how to adapt our method to this case in the paper, and that this leads to significant improvements, inline with previous work.
>
> SOTA: AIS (Neal, 2001), also known as the Jarzynski-Crooks equality in physics (Jarzynski, 1997; Crooks, 1998), is indeed the gold standard for normalizing constant estimation, with a huge body of literature devoted to the problem - those three papers combined still receive over 1,000 citations per year. Note that Metropolis corrected versions of AIS require using high variance REINFORCE type estimators for end-to-end gradients, and therefore make learning sampling parameters very difficult (Thin et al., 2021). Among the unadjusted AIS variants, indeed the recently proposed UHA (Geffner and Domke, 2021; Zhang et al., 2021), which itself is based on ULA (Wu et al., 2020; Thin et al., 2021) are SOTA to our knowledge.
>
> Numerical stability: The reviewer makes an excellent point. As we describe in our Limitations Section 5, we have found in our experiments that numerical stability can be a problem. We first would like to argue that our proposed scheme drastically reduces the variance of the AIS estimator. This naturally translates into more stable gradients when differentiating through the forward sampling paths. See e.g. Table 3 where MCD-UHA drastically reduces the error bars (supposedly from unstable chains) of the UHA estimator. Second, it is a well known problem that differentiating through long sampling paths leads can be unstable. It is out of the scope of our work to investigate detailed solutions to this problem, but we have e.g. observed that placing a stop_gradient around $\nabla_x  \log \pi(x)$ leads to improvements (albeit at the expense of losing some of our theoretical underpinnings).
>
> Bold face results: We did not highlight any particular (e.g. the best) results in Tables 1&2 as we would like to invite the reader to not only look for the best performing setup, but also e.g. notice the relative performance between using different numbers of steps, with or without MCD, etc. For example, MCD-ULA with a ¼ the number of sampler steps still outperforms ULA in Table 1.

---

> > ### Comment · Reviewer_KR2v · 2022-08-09
> > **Thank you for your reply**
> >
> > Dear author, I appreciate the clarification on amortized inference and the SOTA result reassurance. Your paper looks promising, and I will raise my score.

---

### Meta-Review · Area_Chair_gFz5 · 2022-08-26

**Recommendation:** Accept
**Confidence:** Certain

**Metareview:**

The submission presents a novel and interesting method using recent advances in score-based diffusion to improve the recently proposed differentiable AIS log marginal likelihood estimates. The experiments clearly show the benefit of using Monte Carlo diffusion. The writing is clear and of high quality. For these reasons, all reviewers were unanimous in recommending acceptance.

AC notes: (1) out of curiosity, how is the adjusted Langevin and Hamiltonian versions perform on the static targets?, and (2) further to the question below on the timings, would it be useful to add a comparison where the time is kept similar between U(L/H)A and that with MCD, for example, 2K intermediate densities for U(L/H)A and K for the MCD variants.

**Award:**

No

---

### Decision · Program_Chairs · 2022-09-14

Accept